# An efficient approach for broadband sound absorption using periodic multi-layer solid inclusions in acoustic coatings

**Xiaogang Li, Peng Liu, Xilong Zhang, Xiaoyang Zhu, Hongbo Zhang**[ID]*

School of Mechanical and Automotive Engineering, Qingdao University of Technology, Qingdao, China

* zhanghongbo@qut.edu.cn

**Data Availability Statement:** All relevant data are within the manuscript and its Supporting Information files.

## Abstract

Anechoic coatings are of significance for enhancing the acoustic stealth of underwater equipment. Acoustic coatings with multi-layers of periodic solid inclusions are proposed and investigated for broadband sound absorption. Firstly, an analysis model is derived to calculate effective material and geometric parameters of the layer of solid scatterers. Thereafter the acoustic absorption properties can be obtained by the transfer matrix method. Compared with the finite element method, the analytical model is proven to be viable and efficient. The effect of different geometrical parameters on the acoustic performance is investigated when there is one layer of steel inclusions. In addition, the genetic algorithm is used to quickly achieve the broadband absorption for the two-layer case. The optimized structure, featuring subwavelength thicknesses, has significantly improved its sound absorption performance across a wide frequency range spanning from 1780 Hz to 8890 Hz, covering over two octaves. Finally, the influence of different materials of scatterers on the sound absorption is investigated. This study can provide theoretical guidance for the design and optimization of acoustic coatings with multi-layer solid inclusions.

## 1. Introduction

An anechoic coating is a key factor in the stealth of underwater equipment [1–6]. Typically, anechoic coatings are composed of viscoelastic periodic structures that either contain cavities or are filled with solid scatterers. Such periodic structures, commonly known as acoustic metamaterials [7], exhibit a remarkable capability: they enable sub-wavelength acoustic modulation. This unique property, in turn, renders acoustic metamaterials highly versatile, with a diverse array of potential applications spanning aerospace, biomedical devices, and numerous other fields. Consequently, their importance in advancing modern technologies cannot be overstated.

Recently, periodic acoustic coatings with cavities have received widespread attention and research for their exceptional acoustic performance [8–12]. These coatings exhibit good sound absorption by leveraging the monopole resonance of the cavities. The mechanism enhancing the sound absorption performance is that when the frequency of the incident wave is close to

**Funding:** We gratefully acknowledge support in this work from the National Natural Science Foundation of China (Grant No. 12374447 and 11874034) and the Taishan Scholar Program of Shandong (Grant No. ts201712054). The funders had no role in study design, data collection and analysis, decision to publish, or preparation of the manuscript.

**Competing interests:** The authors have declared that no competing interests exist.

that of the monopole resonance frequency of the cavity, the scattering phenomenon is enhanced, which in turn converts the longitudinal wave into the transverse wave, greatly improving the sound absorption characteristics. However, the introduction of cavities will reduce the pressure resistance of viscoelastic materials, such as rubber, to hydrostatic pressure [11, 13]. In addition, the sound-absorbing performance is reduced significantly under high hydrostatic pressure.

Viscoelastic materials with hard inclusions also have special practical importance, as they can improve the pressure bearing characteristics of acoustic coatings and keep the sound absorption characteristics basically unchanged [14]. However, there is much less research on this topic, and there is a lack of an efficient analytical model for broadband sound absorption in acoustic coatings with periodic multi-layer solid inclusions.

Sharma et al. [15, 16] developed an analytical model to study sound transmission through a soft elastic medium with periodic cavities submerged in water. However, the sound absorption bands they obtained were relatively narrow. Fang et al. used effective medium approximation (EMA) and genetic algorithm (GA) optimization to achieve broadband underwater absorption in the multi-layer metagratings [8]. In these studies, the transfer matrix method (TMM) played a pivotal role in modeling and analyzing the sound absorption characteristics of these complex structures. The TMM is a fast and simple method that is applicable in various physics fields, such as acoustics, quantum mechanics, and optics, to analyze wave propagation in one-dimensional systems [17, 18].

Wen et al. used the finite element method (FEM) to investigate underwater acoustic absorption of the viscoelastic materials embedded with local resonance scatterers [19, 20]. It should be noted that the FEM offer precise modeling of non-homogeneous materials, but it demands significant computational resources and are unsuitable for high-frequency simulations of large-scale material structures.

Due to the complicated interactions, there is a lack of analytical solutions for acoustic coatings with multi-layer solid inclusions. For such soft materials with hard inclusions, they can be analyzed using the effective frequency-dependent dynamic density and the effective stiffness [14]. This analysis is similar to the Johnson-Champoux-Allard (JCA) model for the porous materials with effective density and effective bulk modulus [21–26]. There are also many research papers about porous matrix with periodic gratings of hard inclusions [27–29] and acoustic metamaterials [30, 31], which provide important references for the interaction mechanism between acoustic coatings and incident acoustic waves. Besides, periodic viscoelastic materials with hard inclusions can lead to dipole resonances which are core factors affecting the acoustic characteristics of the acoustic coatings.

The above related studies either contain cavities that weaken the pressure-bearing capacity or focus solely on finite element analyses. Although some articles achieve low-frequency broadband sound absorption, the structures they employ are relatively complex.

To enhance the pressure-bearing capabilities of acoustic coatings, periodic cavities are substituted with periodic solid inclusions. The main contribution of this article is to improve the low-frequency broadband sound absorption of two-layer solid inclusions by using GA and EMA. An optimized structure, consisting of solely two layers of scatterers, has been found to achieve a remarkable average sound absorption coefficient of 0.7906 spanning the frequency from 10 Hz to 10,000 Hz. This demonstrates the exceptional performance of the design in effectively absorbing sound energy within a broad spectrum of audible frequencies.

An analysis model was developed to calculate the effective geometric and material parameters of the layers of solid scatterers under normal incidence. This approach allows treating the periodic viscoelastic medium as a homogeneous medium with effective properties. The analytical results were validated by comparison with those obtained from the FEM [32]. Efficient

optimization of the sound-absorbing coating with multi-layer inclusions, aimed at enhancing sound absorption performance, was carried out using the TMM and the GA [33]. This study offers theoretical insights for the design and optimization of acoustic coatings featuring periodic multi-layer hard inclusions.

## 2. Model and calculation methods

The two-dimensional cross section of a polydimethylsiloxane (PDMS) medium with two layers of periodic cylindrical inclusions is depicted in Fig 1. Due to the periodicity of hard inclusions, this structure is infinite along the $y$ and $z$ directions. From left to right, the corresponding materials are half infinite water, PDMS, steel, and half infinite air. The material parameters of fluids are $\rho_w = 1000$ kg/m$^3$(density), $c_w = 1500$m/s (sound speed) for water, and $\rho_a = 1.2$ kg/m$^3$, $c_a = 340$m/s for air. The material parameters of solids are given in Table 1. As shown in Fig 1, the radii of the hard inclusion are $r_1$ and $r_2$. The distances between adjacent inclusions are $a_1$ and $a_2$. The thicknesses of the PDMS layer and steel layer are $t = t_1 + t_2$ and $t_s$, respectively.

The analytical model for determining the acoustic properties of an array of rigid inclusions within an elastic matrix is formulated based on the dynamic behavior of the inclusions triggered by an incoming acoustic wave. The fundamental equation governing this process can be found in various sources [14, 34–36]. For a solid layer composed of solid scatterers, it can be treated as a homogenized layer, characterized by effective thickness $t_e$, effective density $\rho_e$, and effective longitudinal modulus $\kappa_e$.

In this article, the effective thickness $t_e$ can be calculated by [14, 37]

$$t_e = \frac{2a}{\pi} \ln \sec\left(\frac{\pi r}{a}\right) \tag{1}$$

In order to calculate effective density $\rho_e$ and effective longitudinal modulus $\kappa_e$, a two-step homogenization model is used. The global effective parameters of the cylinders are first calculated. Then, based on these parameters, the effective parameters of the solid layer of scatterers are derived.

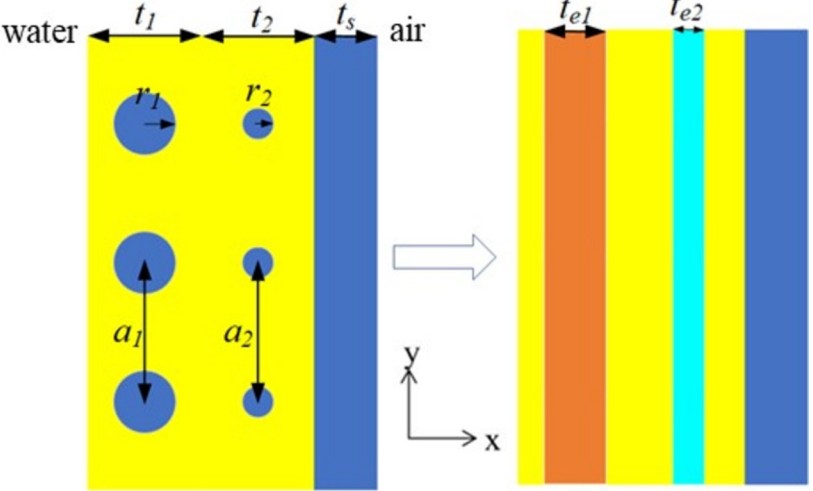

**Fig 1. Cross section of the metagrating and the distribution of effective homogeneous layers.** Materials along the $x$ direction are water, viscoelastic PDMS plate with periodic solid scatterers, steel plate, and air.

**Table 1. The solid material attributes.**

| Solids | Density ($\rho$/kg·m$^{-3}$) | Young's modulus (E/GPa) | Poisson's ratio ($v$) | Loss factor ($\eta$) |
|---|---|---|---|---|
| Aluminum | 2730 | 77.6 | 0.35 | —— |
| Steel | 7890 | 210 | 0.3 | —— |
| Lead | 11,600 | 40.8 | 0.37 | —— |
| Tungsten | 19,250 | 411.0 | 0.28 | —— |
| PDMS | 1000 | $1.8\times10^{-3}$ | 0.4997 | 0.3 |

The global effective stiffness $\kappa_g$ of the cylinders in the PDMS medium averaged over the unit cell is given as [38]

$$\kappa_g = \frac{\kappa\kappa_s}{\alpha_1\kappa + \kappa_s - \alpha_1\kappa_s} \tag{2}$$

in which $\kappa$ is the longitudinal modulus of the PDMS material, $\kappa_s$ is the longitudinal modulus of the hard inclusion, and $\alpha_1 = \pi r^2/a^2$ is the filling fraction.

The global effective density $\rho_g$ can be computed as [34]

$$\rho_g = \rho_{g1} - i\rho_{g2} \tag{3}$$

in which

$$\frac{\rho_{g1}}{\rho} = 1 + \alpha(\delta - 1)\frac{\left((\omega_0/\omega)^2 - (1+\beta)/(\delta+\beta)\left((\omega_0/\omega)^2 - 1\right)\right) + (\psi/\omega)^2}{\left((\omega_0/\omega)^2 - 1\right)^2 + (\psi/\omega)^2} \tag{4}$$

$$\frac{\rho_{g2}}{\rho} = \frac{\alpha\psi(\delta - 1)^2}{\omega(\delta+\beta)\left(\left((\omega_0/\omega)^2 - 1\right)^2 + (\psi/\omega)^2\right)} \tag{5}$$

in which $\delta = \rho_s/\rho$, $\beta = \frac{1+\alpha_1}{1-\alpha_1}$, $\omega_0 = \gamma\frac{c_s}{r}$ is the dipole resonance angular frequency with

$$\gamma = \left(\frac{8(1+\alpha_1{}^2)}{(\delta+\beta)((1+\alpha_1{}^2)\ln(1/\alpha_1) + 1 - \alpha_1{}^2)}\right)^{0.5} \tag{6}$$

$c_s = \sqrt{\mu/\rho}$ is the shear wave speed in the host medium and $\psi = \frac{\omega}{\delta+\beta}\text{imag}(H)$ in which "imag" stands for deriving the imaginary part of a complex value, and $H$ is given as [14, 39]

$$H = 1 - \frac{N}{D} \tag{7}$$

$$N = 2\tau^2[\boldsymbol{\Omega}_{00}(\tau,\chi) - \boldsymbol{\Omega}_{00}(\chi,\tau)] - 4\tau[\boldsymbol{\Omega}_{10}(\tau,\chi) - \boldsymbol{\Omega}_{01}(\chi,\tau)] + 4\tau\sqrt{\alpha_1}[\boldsymbol{\Omega}_{01}(\tau,\chi) - \boldsymbol{\Omega}_{10}(\chi,\tau)] \\ - 8\sqrt{\alpha_1}[\boldsymbol{\Omega}_{11}(\tau,\chi) - \boldsymbol{\Omega}_{11}(\chi,\tau)] \tag{8}$$

$$D = \tau^2(1-\alpha_1)[\boldsymbol{\Omega}_{00}(\tau,\chi) - \boldsymbol{\Omega}_{00}(\chi,\tau)] - 2\tau\sqrt{\alpha_1}[\boldsymbol{\Omega}_{01}(\tau,\chi) - \boldsymbol{\Omega}_{10}(\chi,\chi) + \boldsymbol{\Omega}_{10}(\chi,\tau) - \boldsymbol{\Omega}_{00}(\chi,\chi)] \\ + 2\tau\alpha_1[\boldsymbol{\Omega}_{01}(\chi,\tau) - \boldsymbol{\Omega}_{01}(\tau,\tau) + \boldsymbol{\Omega}_{10}(\tau,\chi) - \boldsymbol{\Omega}_{10}(\tau,\tau)] \tag{9}$$

in which $\tau = \sqrt{i\sigma}$, $\chi = \sqrt{i\sigma/\alpha_1}$, $\sigma = \omega a^2/v$, $v$ stands for kinematic viscosity $\boldsymbol{\Omega}_{mn}(\tau,\chi) = I_m(\tau)K_n(\chi)$ and $I_m$ ($K_n$) is the first (second) kind, $m$ ($n$) order modified Bessel function.

Thereafter the effective density $\rho_e$ and the effective longitudinal modulus $\kappa_e$ can be described as [40]

$$\rho_e = \frac{\rho_g + \rho(\alpha_e - 1)}{\alpha_e} \tag{10}$$

$$\kappa_e = \frac{\kappa \kappa_g \alpha_e}{\kappa - \kappa_g + \alpha_e \kappa_g} \tag{11}$$

in which $\alpha_e = t_e/a$.

The effective impedance $Z_e$ and the effective wave number can be obtained as [8, 15]

$$Z_e = \sqrt{\rho_e \kappa_e} \tag{12}$$

$$k_e = \omega \sqrt{\rho_e/\kappa_e} \tag{13}$$

Eqs 12 and 13 can also be used to obtain the impedance and wave number of the homogenous layer. Based on these wave numbers, impedances, and thicknesses, the sound absorption spectrum of the multilayer structures can be obtained using the TMM [8, 15]. The final expression for the coefficient $\alpha$ is

$$\alpha = 1 - \frac{Z_a}{Z_w}|t_p|^2 - |r_p|^2 \tag{14}$$

where $Z_a$, $Z_w$, $t_p$, and $r_p$ stand for water impedance, air impedance, transmission, and reflection coefficients, respectively.

It should be noted that certain conditions must be met to apply this analytical method. First, the density of the hard inclusions must be significantly higher than that of the surrounding elastic medium [37], as Eq (1) is derived by treating the scatterer as a rigid body. Second, the longitudinal sound speed of the host elastic medium should be much greater than its shear wave speed. Third, the analytical model used in this study is suitable for sound wavelengths that are significantly larger than both the inclusion radius $r$ and the spacing $a$. In our research, the scatterer's radius is always smaller than the lattice constant, ensuring that the medium can be considered homogeneous, which requires $\omega_0 a/c_l \ll 1 \left(c_l = \sqrt{\kappa/\rho}\right)$ and $\omega_{max} a/c_l << 1$ ($\omega_{max} = 2\pi f_{max}$, $f_{max}$ represents the upper frequency limit of the study). As shown in Eq (6), when the filling fraction is either extremely small or extremely large, the condition $\omega_0 a/c_l << 1$ may not be satisfied. In such cases where the analytical model does not meet these conditions, the results are replaced by those calculated using the FEM.

To verify the correctness of the analytical method, the FEM is also used to calculate the acoustic properties [14, 32]. A simulation in the frequency domain by using the COMSOL Multiphysics software is carried out. Only one unit cell is considered due to the periodicity. As shown in Fig 2, the water and air domains are imitated by the acoustic module governed by Helmholtz equation, and the PDMS domain and steel domain are simulated by the solid mechanical module regulated by the Navier equations. Periodic boundary conditions are applied to the upper and lower boundaries, while the perfect matching layers are used to eliminate reflections on the left and right boundaries. The structural-acoustic interaction is adopted between the fluid domain and solid domain. The background pressure field (water domain) simulates the incident acoustic wave with unity pressure amplitude. The entire region is divided into triangular meshes, the size of which is controlled by the physical field. The

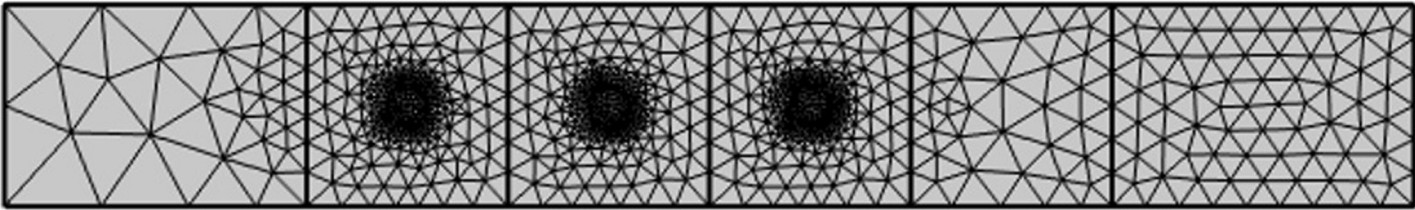

**Fig 2. The FEM mesh model for the three-layer case, where 7919 degrees of freedom are used.**

reflected and transmitted pressures can be obtained at the interface between the fluid and solid media on the incidence and transmission sides, respectively [36].

## 3. Results and discussions

To ensure high accuracy, 7919 degrees of freedom are utilized for the three-layer case in Fig 2. This is achieved through the utilization of the Refine Mesh Technique (RMT), as illustrated in Fig 3, thereby ensuring mesh convergence without redundancy. To validate the predictive model of acoustic coatings, we compare the absorption coefficients derived from the current simulation method (FEM) with those obtained by Sharma et al. [13]. As shown in Fig 4, comparing our results (blue lines) with those results (dotted lines) from reference [13], we can see that our simulated results agree reasonably with those of the reference over the entire frequency region, thus validating the present model.

To validate the analytical method (TMM) utilized in this paper, results from the FEM are also presented in Fig 5 for comparison. The geometrical parameters for the models are as follows: $t = 60$ mm, $t_s = 20$ mm, $r = 2$ mm, $a = 20$ mm for Fig 5(A); $t_1 = t_2 = 30$ mm, with other parameters being the same as those in Fig 5(A), for Fig 5(B); and $t_1 = t_2 = t_3 = 20$ mm, with other parameters being the same as those in the previous two figures, for Fig 5(C).

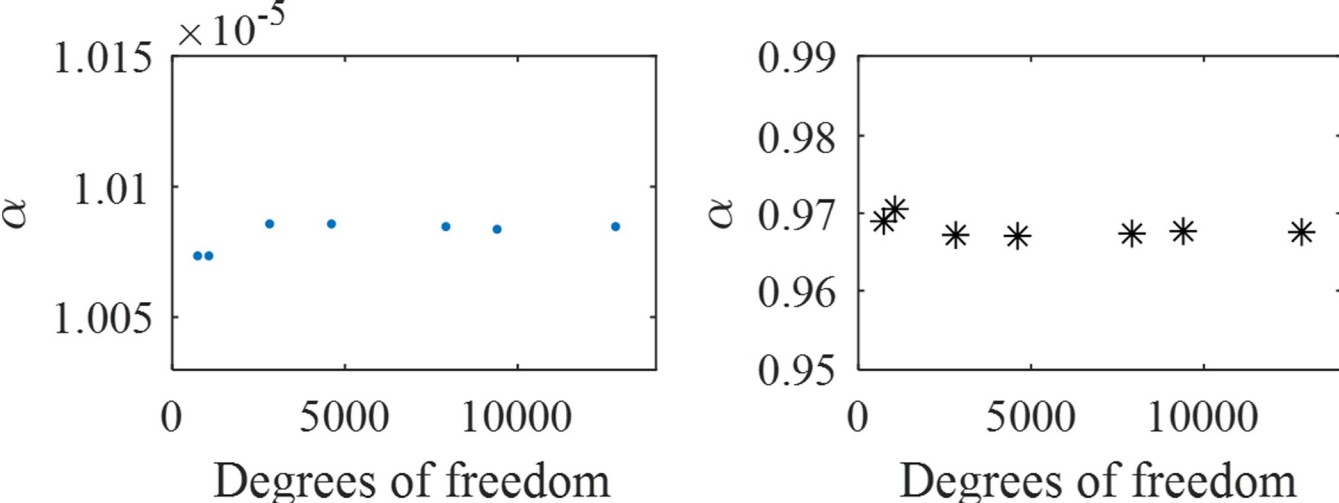

**Fig 3. The RMT diagram for an acoustic coating with three-layer periodic inclusions, in which the start (10 Hz, blue dots) and cut-off (10000 Hz, black asterisks) frequencies of sound absorption coefficients are distinguished.** Notably, beyond 2818 degrees of freedom, the sound absorption remains almost unaltered as the number of degrees of freedom increases, which means that the number of degrees of freedom does not significantly affect the sound absorption performance.

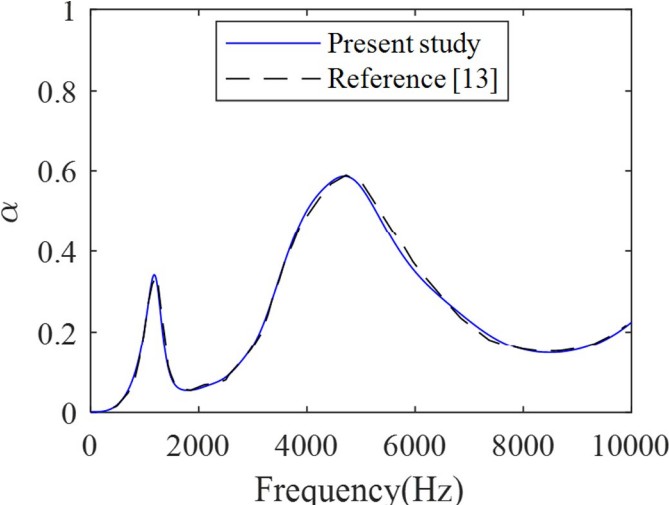

**Fig 4. A comparison of the absorption coefficients obtained from the current simulation method (FEM) with those obtained by Sharma et al. [13].**

From Fig 5(A) and 5(B), we can see that the results calculated by TMM are in good agreement with those calculated by FEM. For Fig 5(C), up to 5380 Hz, the results calculated by the two methods are quite consistent. However, above 5380 Hz, there are significant differences in the results, yet the overall trend remains consistent. From the above calculation results, it can be seen that the results of the one-layer case and the two-layer case are reliable, while the results of the three-layer case can also meet the requirements at medium and low frequencies ($\omega_{max}\, a/c_l \leq 0.5$), with significant differences at high frequencies, but the trend remains consistent. To ascertain the accuracy of the TMM, we calculate the relative error $(\alpha_t\text{-}\alpha_f)/\alpha_f$, where $\alpha_t$ represents the sound absorption coefficient obtained using the TMM, while $\alpha_f$ represents the sound absorption coefficient obtained using the FEM. Within the frequency range of 10Hz to 10kHz, which is of our concern, the relative error for one-layer case is only 1.8%, while for two-layer case, it is 2.7%. Even for the three-layer case, which seems to have a larger error, the relative error is only 6.02%. As can be seen from the above results, the results calculated using the TMM are highly credible.

In order to improve the absorption performance of the anechoic coating and reveal the underlying mechanism, different geometrical parameters are investigated with one layer of steel inclusions. First, the influence of the thickness of the PDMS on the absorption performance is shown in Fig 6(A), with other parameters remaining the same as those in Fig 5(A). We can see from Fig 6(A) that increasing the thickness of the PDMS from 15 mm to 60 mm enhances the absorption performance in the low frequency range. Especially when increasing from 15 mm to 30 mm, the improvement in sound absorption efficiency at low frequencies is most significant. Meanwhile the absorption performance at high frequencies (above 7020 Hz) is reduced, which means that the PDMS is applied as a resonant part and loses partial absorption ability in high frequencies. When the thickness reaches 60 mm, the sound absorption performance of the concerned frequency range is greatly improved.

The influence of the distance $a$ between the inclusions on the absorption performance is also studied in Fig 6(B), with all other parameters being the same as in Fig 5(A). From Fig 6(B), we can see that the influence of the distance $a$ is small as a whole. Increasing the distance between inclusions will slightly reduce the sound absorption performance. These results suggest that as the distance $a$ between the inclusions increases, the interaction between the

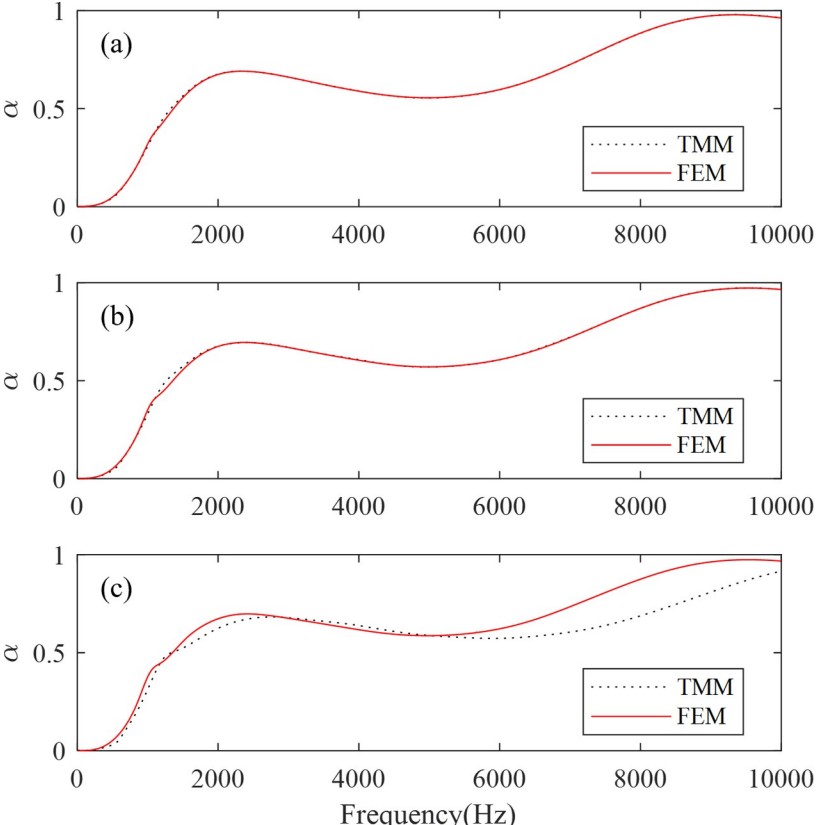

**Fig 5. Sound absorption spectrum calculated by TMM (dots) and FEM (solid lines).** (a) one layer of hard inclusions, (b) two layers of hard inclusions, (c) three layers of hard inclusions.

incident sound wave and the periodic scatterers diminishes, leading to a decrease in both dissipation and sound absorption performance.

The influence of the radius $r$ of the inclusions on the absorption is depicted in Fig 6(C), with other parameters remaining the same as those in Fig 5(A). It can be seen that the radius significantly and complexly affects the absorption performance. This can be explained as follows: From Eqs (3)–(5) and (10), it can be deduced that the dipole resonance frequency affects the equivalent density, which in turn influences the sound absorption coefficient. Similarly, based on Eqs (2) and (11), and $\alpha_1 = \pi r^2 / a^2$, it is evident that the radius impacts the equivalent modulus, which further affects the sound absorption coefficient. Increasing the radius from 2 mm to 6 mm enhances sound absorption performance at low to medium frequencies while reducing it at high frequencies. However, when the radius is increased from 6 mm to 8 mm, sound absorption performance decreases at low and high frequencies but significantly improves at intermediate frequencies.

From the above research, it can be seen that the main geometric parameters affecting sound absorption performance of the one-layer case are the radius of the scatterer and the thickness of the PDMS layer. Compared to the influence of the radius and the thickness, the distance between the inclusions is clearly a secondary factor. It also reveals that the sound absorption of the proposed structures can be flexibly modulated by adjusting the geometric parameters mentioned above at low frequencies, which is of great importance for the acoustic stealth performance of underwater equipment.

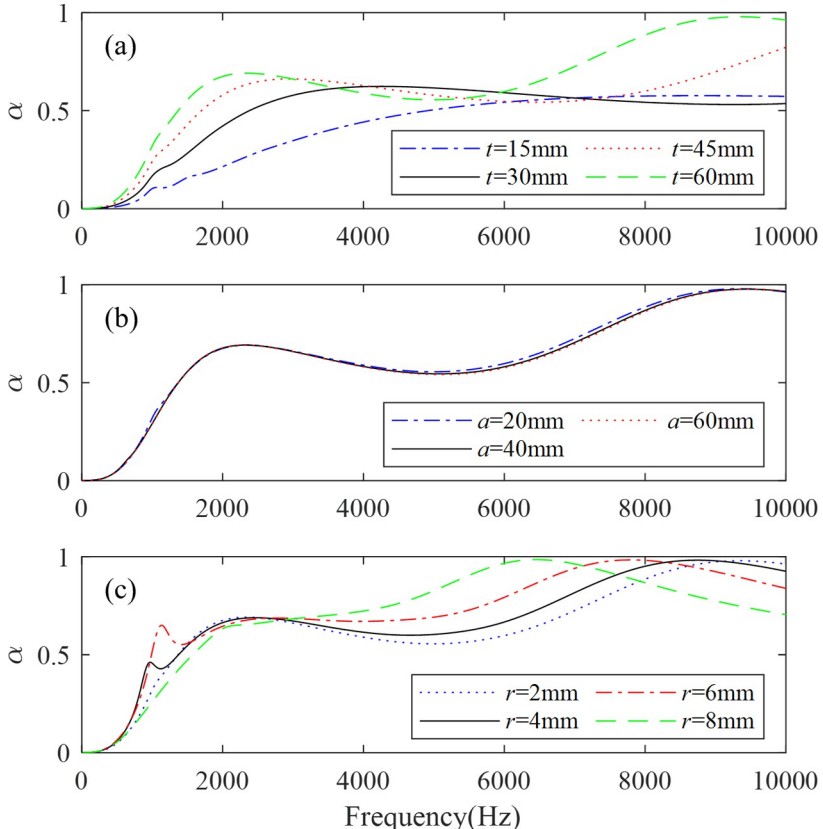

**Fig 6.** The influences of (a) the thickness $t$ of the PDMS, (b) the distance $a$ between the inclusions, and (c) the radius $r$ of the inclusions on the sound absorption coefficients of the anechoic coatings with one layer of steel inclusions.

The geometric parameters of acoustic coatings significantly impact the absorption coefficient, usually exhibiting a nonlinear relationship. Hence, directly achieving broadband and superior sound absorption performance through experience or parametric scanning poses significant challenges. Various optimization Algorithms, including Nelder-Mead method, GA, Simulated Annealing (SA), and Particle Swarm Optimization (PSO), have garnered widespread adoption in acoustic applications [41–44]. To achieve enhanced absorption performance within the specified frequency range, we employ a global, fast and robust GA [22, 23] for optimization. The assumed total thickness of the structure is 60 mm. The parameters that are primarily optimized involve $r_1$, $r_2$, $a_1=a_2=a$, $t_1$, $t_2$. The definition of the optimization vector and constraint range for these parameters is as follows

$$\vec{x} = [r_1, r_2, a, t_1, t_2] \tag{15}$$

where $r_1 \in [1,9]$mm, $r_2 \in [1,9]$mm, $a \in [5,60]$mm, $t_1 \in [10,50]$mm, $t_2 \in [10,50]$mm.

The determination of the objective function involves finding the maximum value of the average absorption coefficients within a designated frequency range ($f_{min}$, $f_{max}$), This

maximum value is denoted as

$$\text{Max} : \{\alpha\} = \frac{\displaystyle\sum_{f_{\min}}^{f_{\max}} \alpha(f_i, \vec{x})}{\Delta f} \qquad (16)$$

where $\Delta f = f_{\min}\text{-}f_{\max}$.

The optimized results are compared before and after optimization in Fig 7. The material and geometrical properties before optimization are the same as those in Fig 5(B). Optimized geometrical parameters are as follows: $r_1 = 4.5$ mm, $r_2 = 2.5$ mm, $a_1 = a_2 = 10.4$ mm, $t_1 = 43.4$ mm, $t_2 = 16.6$ mm. The reference case is the same as the optimized case but $a = 20$ mm. It is worth pointing out that our personal computer, with an Intel® Core™ i7-8700 CPU @ 3.20GHz and 16 GB of RAM, can obtain optimized results in about 4 minutes. In contrast, joint simulation calculations using COMSOL and MATLAB are very slow and, in most cases, fail to produce results. As can be seen from Fig 7(A), the optimized structure has significantly improved its sound absorption performance within the wide frequency range of 1780 Hz to 8890 Hz (the corresponding wavelengths are about 0.84 m to 0.17m), which means that this is a subwavelength absorber (the total thickness of the acoustic coating is 0.08 m). The only disadvantage is that the optimized structure will reduce the sound absorption performance in the lower frequency range. It should be pointed out that the objective function involves finding the maximum value of the average absorption coefficient within the frequency range (from 0 to 10000Hz). Therefore, to optimize the low-frequency sound absorption performance, the optimization frequency range can be changed (such as from 10 Hz to 1000 Hz). According to the previous discussion, the influence of spacing on single-layer sound absorption is very small, but the distance between hard inclusions of two-layer structures on the sound

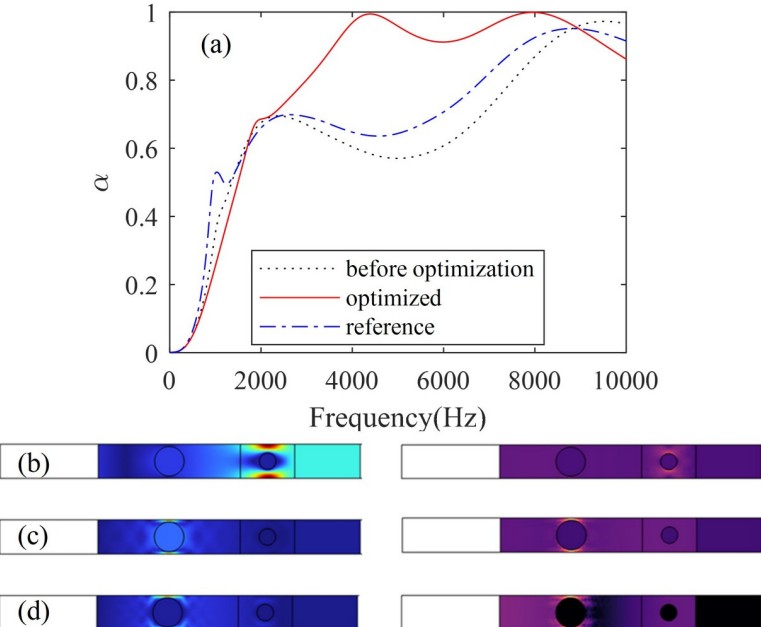

**Fig 7.** The optimized sound absorption coefficient (a) and the displacement field (left side) and the energy dissipation density (right side) at the peak frequencies 1960Hz (b), 4340Hz (c) and 7960Hz (d) of the anechoic coating with two layers of steel inclusions.

absorption performance is still significant, which indicates that there is some coupling in the double-layer structure.

Despite the numerous advantages of GA, it also possesses a significant number of drawbacks, including the computational cost associated with large search spaces, the sensitivity to parameter settings, and the potential for getting stuck in local optima. We also acknowledge that while GA has proven effective in our study, there may be other optimization techniques that could offer advantages in certain scenarios.

To further understand the underlying mechanism of broadband sound absorption, the displacement field and the energy dissipation density of the sound-absorbing structure are studied at the peak frequencies of 1960Hz ($\alpha = 0.684$), 4340Hz ($\alpha = 0.994$) and 7960Hz ($\alpha = 0.999$), respectively. As shown in Fig 7(B)–7(D), the acoustic wave is incident normally from the left side of the structure. The displacement field diagram is on the left side, while the energy dissipation diagram is on the right side. Overall, the displacement amplitude is large, and the corresponding level of energy dissipation is also high. When the excitation frequency is 1960Hz, the maximum displacement amplitude is located in the upper and lower PDMS of the right scatterer, and the corresponding energy dissipation is concentrated in these two places. At the second peak (4340 Hz), the areas with the highest displacement amplitude are the PDMS above and below the left scatterer, in which energy dissipation is also concentrated. At the third peak (7960 Hz), energy dissipation is also concentrated above and below the left scatterer. However, for the third peak, the dissipation of energy occurs closer to the incident side compared with the second peak. From these phenomena, we can observe that lower frequencies are more difficult to dissipate, and their dissipation positions are closer to the steel plate side. As the frequency increases, energy becomes easier to dissipate and occurs closer to the incident side. In addition to the fact that the incident longitudinal waves are converted into transverse waves under the influence of the periodic cylinders and subsequently dissipated by the PDMS [14], another important reason is that, the acoustic impedance at the incident side is relatively matched, while, at the steel plate side, the acoustic impedance is mismatched, causing further dissipation. We can provide a quantitative explanation for this from Fig 8. As evident from Fig 8, at 1960 Hz, the normalized surface acoustic resistance $Z_r$ is 2.56, and the normalized surface acoustic reactance $Z_i$ is -1.52, whereas at 4340 Hz, these values are 1.04 and 0.16, respectively,

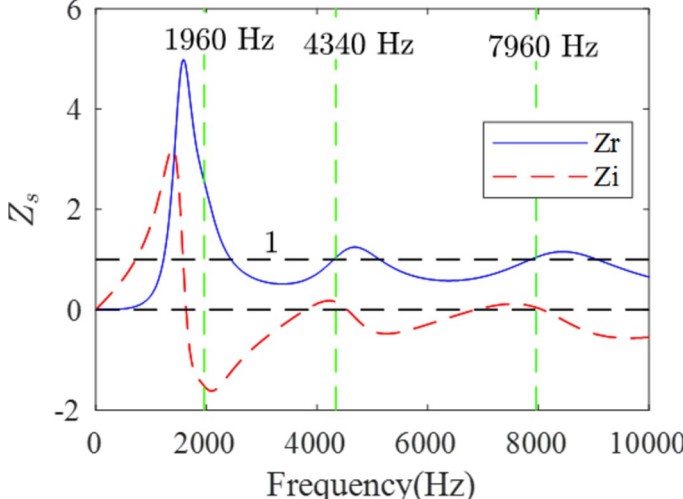

**Fig 8. The normalized acoustic resistance (blue lines) and reactance (red dotted lines) of the optimization case in Fig 7.**

and at 7960 Hz, they are 1.05 and 0.05. It is well-known that the requirements for perfect sound absorption are met when $Z_r = 1$ and $Z_i = 0$. This explains why near-perfect sound absorption is achieved at 4340 Hz and 7960 Hz, whereas at 1960 Hz, the corresponding conditions are not met, resulting in a sound absorption coefficient of only 0.684. Furthermore, as seen in Fig 8, the normalized surface acoustic impedance within the range from 1960 Hz to 10 kHz approaches the conditions for perfect sound absorption, enabling efficient broadband sound absorption within this frequency range.

To select suitable materials for better sound absorption, different materials of scatterers are compared. The properties of studied materials are listed in Table 1. As can be seen from Fig 9 at low frequencies (10Hz to 1520Hz), Tungsten exhibits the best sound absorption performance, while Lead performs best from 1520Hz to 3950Hz, steel from 3950Hz to 9090Hz, and aluminum has the superior sound absorption performance after 9090Hz. In the studied frequency range, the average sound absorption coefficients, ordered from small to large are: 0.7095 (Aluminum), 0.7462 (Tungsten), 0.7810 (Lead), 0.7906 (Steel). Consequently, when applying these materials in specific applications, it is crucial to select the suitable material based on the frequency range and the desired sound absorption performance.

Our future research plan is to develop a multifunctional acoustic metamaterial. Firstly, the acoustic metamaterial will incorporate carbon nanotubes or other materials into its viscoelastic matrix material, enhancing its acoustic pressure resistance and structural strength. Secondly, by substituting simple scatterers with local resonant structures and incorporating cavities, this hybrid design aims to achieve lower frequency and broadband sound absorption effects. Third, we plan to utilize natural or recycled materials to reduce costs and minimize the impact on underwater ecology. Fourth, utilize advanced technologies, including 3D printing, to efficiently produce high-quality experimental samples. Finally, the final simulation results will be verified through experiments.

In addition, we will provide more sensitivity analysis to cover a wider range of variables. Factors such as environmental conditions, scale effects, material inhomogeneity, multi-directional loading, thermal shock loading [45], can significantly affect the functionality and reliability of these materials. This extension of sensitivity analysis is interdisciplinary, offering greater insight into the robustness and elasticity of material designs. It also improves the accuracy of predicting the performance of these materials in a variety of real-world scenarios. We

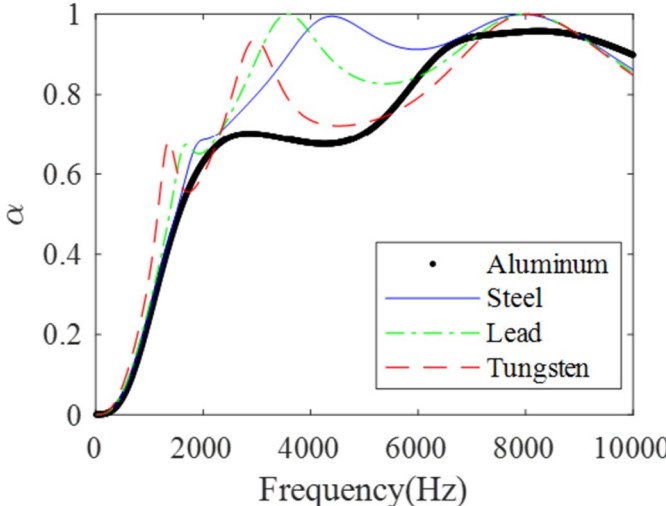

**Fig 9. The influence of different materials of scatterers on the sound absorption.**

will also discuss the prioritization method for the selected variables, providing reasons for including or excluding certain factors.

## 4. Conclusion

In summary, the acoustic performance of acoustic coatings with multi-layers of solid inclusions has been analytically and numerically examined. The analytical results for the one-layer and two-layer cases are reliable, while the analytical results for the three-layer case are accurate at low and medium frequencies, but show significant differences at high frequencies, although the overall trend remains consistent. To improve the absorption performance of the anechoic coating and reveal the underlying mechanism, the influence of the geometrical parameters of a coating with one layer of steel inclusions on the sound absorption coefficient was investigated. The results indicate that the thickness of the PDMS and the radius of the hard inclusions greatly impact sound absorption performance, while the distance between inclusions has little effect. With the help of a genetic algorithm, broadband absorption was quickly achieved (in about four minutes) for the two-layer case. Unlike previous studies, the optimized structure, consisting of just two layers of scatterers, significantly improved sound absorption performance across a wide frequency range of 1780 Hz to 8890 Hz (corresponding to wavelengths of approximately 0.84 m to 0.17 m). This indicates that it functions as a subwavelength absorber, with a total coating thickness of 0.08 m. This study provides theoretical guidance for the design and optimization of acoustic coatings with multi-layer solid inclusions.

## Supporting information

**S1 File. Data for Figs 3–9.**
(ZIP)

**S2 File. Simulation sample.**
(ZIP)

## Author Contributions

**Conceptualization:** Hongbo Zhang.

**Data curation:** Xiaogang Li.

**Formal analysis:** Xiaogang Li.

**Investigation:** Xiaogang Li, Peng Liu, Xilong Zhang.

**Methodology:** Hongbo Zhang.

**Resources:** Xiaoyang Zhu, Hongbo Zhang.

**Software:** Xiaogang Li.

**Supervision:** Peng Liu, Xilong Zhang.

**Validation:** Xiaoyang Zhu.

**Writing – original draft:** Xiaogang Li, Hongbo Zhang.

**Writing – review & editing:** Peng Liu, Xilong Zhang, Xiaoyang Zhu, Hongbo Zhang.

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
