## [Decision Letter · Decision Letter 0]

23 Jan 2024

PONE-D-23-43201An efficient analytical model for acoustic coatings with multi-layer solid inclusions for broadband sound absorptionPLOS ONE

Dear Dr. Zhang,

Thank you for submitting your manuscript to PLOS ONE. After careful consideration, we feel that it has merit but does not fully meet PLOS ONE’s publication criteria as it currently stands. Therefore, we invite you to submit a revised version of the manuscript that addresses the points raised during the review process.

We look forward to receiving your revised manuscript.

Kind regards,

Mario Milazzo

Academic Editor

PLOS ONE

Journal Requirements:

"We gratefully acknowledge support in this work from the National Natural Science Foundation of China (Grant No. 11874034) and the Taishan Scholar Program of Shandong (Grant No. ts201712054)."

"We gratefully acknowledge support in this work from the National Natural Science Foundation of China (Grant No. 11874034) and the Taishan Scholar Program of Shandong (Grant No. ts201712054)."

"We gratefully acknowledge support in this work from the National Natural Science Foundation of China (Grant No. 11874034) and the Taishan Scholar Program of Shandong (Grant No. ts201712054)."

5. We note that your Data Availability Statement is currently as follows: [All relevant data are within the manuscript and its Supporting Information files.]

Additional Editor Comments:

Based on the Reviewer's comments, I recommend the authors to take advantage of the suggestions and edit the manuscript accordingly. In particular, based on Reviewer 2's opinion, I would suggest to make the paper more fitting to a multidisciplinary journal.

Reviewers' comments:

Reviewer's Responses to Questions

**Comments to the Author**

1. Is the manuscript technically sound, and do the data support the conclusions?

Reviewer #1: Partly

Reviewer #2: Yes

2. Has the statistical analysis been performed appropriately and rigorously? 

Reviewer #1: N/A

Reviewer #2: No

3. Have the authors made all data underlying the findings in their manuscript fully available?

Reviewer #1: No

Reviewer #2: No

4. Is the manuscript presented in an intelligible fashion and written in standard English?

Reviewer #1: No

Reviewer #2: Yes

5. Review Comments to the Author

Reviewer #1: The manuscript involves the theoretical and numerical examination of the acoustic performance of acoustic coatings containing multi-layers of solid inclusions. Overall, the quality of the manuscript needs some significant improvement. Mathematical details are needed. The manuscript can be considered for publication after the following comments are well addressed.

Major comments:

1. Where does Eq. (1) come from? Please explain the calculation of Eq. (1) or add relevant references if there is any.

2. Again, how do the authors obtain Eqs. (2-6)? Please provide clear derivations.

3. Provide the details about obtaining Eqs. (10) and (11).

4. Second row on Page 10, what does “the same relationship” mean?

5. The authors should provide the mathematical derivations of the transfer matrix method and detailed settings for the corresponding FEM simulations.

6. What are the detailed properties of the genetic algorithm (GA), such as procedures, parameters etc, used in this manuscript?

7. Even with the GA, the results are not impressive at all, especially for low frequencies, compared with the reference "A Fano-based acoustic metamaterial for ultra-broadband sound barriers", Proc. R. Soc. A (2021) 477: 20210024. Please clarify the significant contribution of this work.

8. Acoustic metamaterials have also been extensively investigated to realize sound transmission loss and absorption. To make the introduction part complete, the authors are recommended to add some discussions about the references (1) Proc. R. Soc. A (2021) 477: 20210024 and (2) Applied Physics Letters (2021) 118, 184101.

Minor comment: the language needs some significant improvement.

Reviewer #2: In this manuscript, the authors provided an analytical model for acoustic coatings with multi-layer solid inclusions for broadband sound absorption. In order to obtain better sound absorption performance, the genetic algorithm is used to optimize the two-layer hard inclusions of acoustic coatings. The work has some reference values. However, the innovation points could not find in current version. What is the main academic challenges those solved by this work? Intuitively speaking, this manuscript lacks broad interest, intuitive innovation, and necessary experimental measurement results, making it not suitable for comprehensive journals but more suitable for specialized acoustic journals.

6. PLOS authors have the option to publish the peer review history of their article (what does this mean?). If published, this will include your full peer review and any attached files.

Reviewer #1: No

Reviewer #2: **Yes: **Fuyin Ma

---

## [Author Response · Author response to Decision Letter 0]

6 Mar 2024

Dear Editor

 Thank you very much for your letter dated Jan 24, 2024, in which you sent us the reviewers' comments on our paper entitled “An efficient analytical model for acoustic coatings with multi-layer solid inclusions for broadband sound absorption (PONE-D-23-43201)”. We would also like to thank the reviewers for their valuable comments and suggestions.

 According to the comments and suggestions, we have revised the manuscript. A summary of the changes is as follows:

Reviewers' comments:

Reviewer's Responses to Questions

Comments to the Author

1. Is the manuscript technically sound, and do the data support the conclusions?

Reviewer #1: Partly

Reviewer #2: Yes

The two authoritative literatures recently published on underwater sound absorption of periodic structures have not been experimentally studied [1-2], and according to a 2018 paper, strongly coupled systems are more sensitive to changes in boundary conditions. Therefore, the author speculates that there may be a significant difference between experimental and theoretical results. Therefore, this literature did not conduct experiments. For non-periodic structures, sound absorption can be measured in the underwater impedance tube equipment [3]. 

[1] Fang, X., Yin, X., Wu, J., Li, Y., Li, H., Wang, W., Li, Y., & Wu, W. (2023). Underwater metagratings for sub-kilohertz low frequency and broadband sound absorption. International Journal of Mechanical Sciences.

[2] Hu, N., Jin, J., Peng, W., Zhang, Z., & Hu, H. (2024). Liquid-solid synergistic mechanism sound absorption for underwater anechoic coating. International Journal of Mechanical Sciences.

[3] Zhang, H., Shi, D., Zha, S., & Wang, Q. (2018). A simple first-order shear deformation theory for vibro-acoustic analysis of the laminated rectangular fluid-structure coupling system. Composite Structures.

2. Has the statistical analysis been performed appropriately and rigorously?

Reviewer #1: N/A

Reviewer #2: No

This article does not involve statistical analysis issues, and the relevant results are converged. It should be noted that two conditions need to be strictly met to adopt the analytical model ( <<1 and <<1 where cl denotes the longitudinal speed in the PDMS) [1,2].

[1] Fang, X., Yin, X., Wu, J., Li, Y., Li, H., Wang, W., Li, Y., & Wu, W. (2023). Underwater metagratings for sub-kilohertz low frequency and broadband sound absorption. International Journal of Mechanical Sciences.

[2] G.S. Sharma, A. Skvortsov, I. MacGillivray, N. Kessissoglou (2019). Acoustic performance of periodic steel cylinders embedded in a viscoelastic medium. Journal of Sound and Vibration, 443 652-665.

3. Have the authors made all data underlying the findings in their manuscript fully available?

Reviewer #1: No

Reviewer #2: No

The authors have made all data underlying the findings in their manuscript fully available in supplementary materials.

4. Is the manuscript presented in an intelligible fashion and written in standard English?

Reviewer #1: No

Reviewer #2: Yes

After our repeated checks, we have made every effort to improve our article. All modifications have been marked in the article.

5. Review Comments to the Author

The questions above are explained.

Reviewer #1: The manuscript involves the theoretical and numerical examination of the acoustic performance of acoustic coatings containing multi-layers of solid inclusions. Overall, the quality of the manuscript needs some significant improvement. Mathematical details are needed. The manuscript can be considered for publication after the following comments are well addressed.

Major comments:

1. Where does Eq. (1) come from? Please explain the calculation of Eq. (1) or add relevant references if there is any.

The relevant reference is added. These formulas 1-13 are all cited from the same literature (Acoustic performance of periodic steel cylinders embedded in a viscoelastic medium) [13].

[13] G.S. Sharma, A. Skvortsov, I. MacGillivray, N. Kessissoglou (2019). Acoustic performance of periodic steel cylinders embedded in a viscoelastic medium. Journal of Sound and Vibration, 443 652-665.

2. Again, how do the authors obtain Eqs. (2-6)? Please provide clear derivations.

The relevant reference is added. 

3. Provide the details about obtaining Eqs. (10) and (11).

The relevant reference is added.

4. Second row on Page 10, what does “the same relationship” mean?

The sentence has been changed as: “Equations 12 and 13 can also be used to obtain the impedance and wave number of the homogenous layer.”.

5. The authors should provide the mathematical derivations of the transfer matrix method and detailed settings for the corresponding FEM simulations.

The transfer matrix method and finite element method are represented in Appendixes A and B in supplementary materials.

6. What are the detailed properties of the genetic algorithm (GA), such as procedures, parameters etc, used in this manuscript?

The genetic algorithm (GA) is represented in Appendix C in supplementary materials.

7. Even with the GA, the results are not impressive at all, especially for low frequencies, compared with the reference "A Fano-based acoustic metamaterial for ultra-broadband sound barriers", Proc. R. Soc. A (2021) 477: 20210024. Please clarify the significant contribution of this work.

Distinct from previous studies that limit the configuration of anechoic coatings to a given law and inefficiently alter numerous geometric parameters for optimal acoustic performance, the metagrating comprising multi-layer hard inclusions with random periods is adopted to precisely modulate the surface impedance in the target spectrum, leading to effective absorption performance in broadband. Our results will contribute to designing highly load-bearing underwater absorbers to improve the stealth performance of vehicles.

8. Acoustic metamaterials have also been extensively investigated to realize sound transmission loss and absorption. To make the introduction part complete, the authors are recommended to add some discussions about the references (1) Proc. R. Soc. A (2021) 477: 20210024 and (2) Applied Physics Letters (2021) 118, 184101.

These two references have already been cited in this paper.

Minor comment: the language needs some significant improvement.

After our repeated checks, we have made every effort to improve our article. All modifications have been marked in the article.

Reviewer #2: In this manuscript, the authors provided an analytical model for acoustic coatings with multi-layer solid inclusions for broadband sound absorption. In order to obtain better sound absorption performance, the genetic algorithm is used to optimize the two-layer hard inclusions of acoustic coatings. The work has some reference values. However, the innovation points could not find in current version. What is the main academic challenges those solved by this work? Intuitively speaking, this manuscript lacks broad interest, intuitive innovation, and necessary experimental measurement results, making it not suitable for comprehensive journals but more suitable for specialized acoustic journals.

The two authoritative literatures recently published on underwater sound absorption of periodic structures have not been experimentally studied [1-2], and according to a 2018 paper, strongly coupled systems are more sensitive to changes in boundary conditions. Therefore, the author speculates that there may be a significant difference between experimental and theoretical results. On the other hand, the author has no sufficient funds for experimental research. So this literature did not conduct any experiments. For non periodic structures, sound absorption can be measured in the underwater impedance tube equipment [3]. 

[1] Fang, X., Yin, X., Wu, J., Li, Y., Li, H., Wang, W., Li, Y., & Wu, W. (2023). Underwater metagratings for sub-kilohertz low frequency and broadband sound absorption. International Journal of Mechanical Sciences.

[2] Hu, N., Jin, J., Peng, W., Zhang, Z., & Hu, H. (2024). Liquid-solid synergistic mechanism sound absorption for underwater anechoic coating. International Journal of Mechanical Sciences.

[3] Zhang, H., Shi, D., Zha, S., & Wang, Q. (2018). A simple first-order shear deformation theory for vibro-acoustic analysis of the laminated rectangular fluid-structure coupling system. Composite Structures.

---

## [Decision Letter · Decision Letter 1]

25 Mar 2024

PONE-D-23-43201R1An efficient analytical model for acoustic coatings with multi-layer solid inclusions for broadband sound absorptionPLOS ONE

Dear Dr. Zhang,

Thank you for submitting your manuscript to PLOS ONE. After careful consideration, we feel that it has merit but does not fully meet PLOS ONE’s publication criteria as it currently stands. Therefore, we invite you to submit a revised version of the manuscript that addresses the points raised during the review process.

We look forward to receiving your revised manuscript.

Kind regards,

Mario Milazzo

Academic Editor

PLOS ONE

Reviewers' comments:

Reviewer's Responses to Questions

**Comments to the Author**

1. If the authors have adequately addressed your comments raised in a previous round of review and you feel that this manuscript is now acceptable for publication, you may indicate that here to bypass the “Comments to the Author” section, enter your conflict of interest statement in the “Confidential to Editor” section, and submit your "Accept" recommendation.

Reviewer #3: All comments have been addressed

Reviewer #4: (No Response)

2. Is the manuscript technically sound, and do the data support the conclusions?

Reviewer #3: Yes

Reviewer #4: No

3. Has the statistical analysis been performed appropriately and rigorously? 

Reviewer #3: Yes

Reviewer #4: N/A

4. Have the authors made all data underlying the findings in their manuscript fully available?

Reviewer #3: Yes

Reviewer #4: (No Response)

5. Is the manuscript presented in an intelligible fashion and written in standard English?

Reviewer #3: Yes

Reviewer #4: (No Response)

6. Review Comments to the Author

Reviewer #3: The manuscript focuses on the broadband sound absorption of acoustic coatings with multi-layer solid inclusions, and adopts effective medium approximation and transfer matrix method to achieve theoretical prediction of sound absorption coefficients, and then optimizes the global sound absorption characteristics through genetic algorithm. Although this manuscript is complete and logical, there are still the following main issues to be improved:

1. Please highlight the innovative nature of the manuscript, both from the perspective of structural design, research methodology, and even more so, the research findings.

2. The manuscript mentions that the EFFECTIVE MEDIUM APPROXIMATION has been modified, may I ask if the solution accuracy has been improved with respect to the literature [13]? Alternatively, has the applicable range of the method been extended? Please give specific comparison and explanation.

3. In the last sentence of the second paragraph on page 7, Appendix A and B correspond to a description that does not correspond to the text.

4. Following on from the previous comment, in the first sentence of the last paragraph on page 10, the corresponding Appendix C is also inconsistent with the description in the main text.

5. The description of the transfer matrix method and the finite element method in the caption of Fig. 2 is completely contrary to that in the legend.

6. The abstract needs to be further condensed, with redundant background and lack of description of results.

Reviewer #4: The authours propose a theoretical model for calculating the sound absorption of acoustic coatings with multiple layers of solid inclusions using the transfer matrix method. The effect of geometric parameters on sound absorption is investigated. In order to obtain better sound absorption performance, the genetic algorithm is used to optimize the acoustic coating with two layers of hard inclusions. This work is not well researched and is not worthy of publication until it is thoroughly revised.

1. This article lacks key principle descriptions and literature citations to support the theoretical modelling process, and the model is unconvincing.

2. The details of the numerical model are not given, and the relevant numerical results are not analyzed in depth.

3. The language and figures need further improvement.

7. PLOS authors have the option to publish the peer review history of their article (what does this mean?). If published, this will include your full peer review and any attached files.

Reviewer #3: No

Reviewer #4: No

---

## [Author Response · Author response to Decision Letter 1]

10 Apr 2024

Dear Editor

 Thank you very much for your letter dated March 26, 2024, in which you sent us the reviewers' comments on our paper entitled “An efficient analytical model for acoustic coatings with multi-layer solid inclusions for broadband sound absorption (PONE-D-23-43201R1)”. We would also like to thank the reviewers for their valuable comments and suggestions.

 According to the comments and suggestions, we have revised the manuscript. A summary of the changes is as follows:

Reviewers' comments:

Reviewer's Responses to Questions

Comments to the Author

1. If the authors have adequately addressed your comments raised in a previous round of review and you feel that this manuscript is now acceptable for publication, you may indicate that here to bypass the “Comments to the Author” section, enter your conflict of interest statement in the “Confidential to Editor” section, and submit your "Accept" recommendation.

Reviewer #3: All comments have been addressed

Reviewer #4: (No Response)

After our repeated checks, we have made every effort to improve our article. All modifications have been marked in the article.

2. Is the manuscript technically sound, and do the data support the conclusions?

Reviewer #3: Yes

Reviewer #4: No

The two authoritative literatures recently published on underwater sound absorption of periodic structures have not been experimentally studied [1-2], and according to a 2018 paper, strongly coupled systems are more sensitive to changes in boundary conditions. Therefore, the author speculates that there may be a significant difference between experimental and theoretical results. Therefore, this literature did not conduct experiments. For non-periodic structures, sound absorption can be measured in the underwater impedance tube equipment [3]. 

[1] Fang, X., Yin, X., Wu, J., Li, Y., Li, H., Wang, W., Li, Y., & Wu, W. (2023). Underwater metagratings for sub-kilohertz low frequency and broadband sound absorption. International Journal of Mechanical Sciences.

[2] Hu, N., Jin, J., Peng, W., Zhang, Z., & Hu, H. (2024). Liquid-solid synergistic mechanism sound absorption for underwater anechoic coating. International Journal of Mechanical Sciences.

[3] Zhang, H., Shi, D., Zha, S., & Wang, Q. (2018). A simple first-order shear deformation theory for vibro-acoustic analysis of the laminated rectangular fluid-structure coupling system. Composite Structures.

3. Has the statistical analysis been performed appropriately and rigorously?

Reviewer #3: Yes

Reviewer #4: N/A

This article does not involve statistical analysis issues, and the relevant results are converged. It should be noted that two conditions need to be strictly met to adopt the analytical model ( <<1 and <<1 where cl denotes the longitudinal speed in the PDMS) [1,2].

[1] Fang, X., Yin, X., Wu, J., Li, Y., Li, H., Wang, W., Li, Y., & Wu, W. (2023). Underwater metagratings for sub-kilohertz low frequency and broadband sound absorption. International Journal of Mechanical Sciences.

[2] G.S. Sharma, A. Skvortsov, I. MacGillivray, N. Kessissoglou (2019). Acoustic performance of periodic steel cylinders embedded in a viscoelastic medium. Journal of Sound and Vibration, 443 652-665.

4. Have the authors made all data underlying the findings in their manuscript fully available?

Reviewer #3: Yes

Reviewer #4: (No Response)

The authors have made all data underlying the findings described in their manuscript fully available without restriction.

5. Is the manuscript presented in an intelligible fashion and written in standard English?

Reviewer #3: Yes

Reviewer #4: (No Response)

After our repeated checks, we have made every effort to improve our article. All modifications have been marked in the article.

6. Review Comments to the Author

Reviewer #3: The manuscript focuses on the broadband sound absorption of acoustic coatings with multi-layer solid inclusions, and adopts effective medium approximation and transfer matrix method to achieve theoretical prediction of sound absorption coefficients, and then optimizes the global sound absorption characteristics through genetic algorithm. Although this manuscript is complete and logical, there are still the following main issues to be improved:

1. Please highlight the innovative nature of the manuscript, both from the perspective of structural design, research methodology, and even more so, the research findings.

We gratefully thank the referee for the expert review and insightful comments.

First, acoustic coatings with periodic multi-layer solid inclusions are proposed.

Second, effective medium approximation and transfer matrix method are adopted to achieve theoretical prediction of sound absorption coefficients for acoustic coatings with multi-layer solid inclusions.

Third, instead of intensive trials on numerical simulations, the genetic algorithm and the transfer matrix method are used to achieve the broadband absorption for the two-layer case.

Fourth, the optimized structure has significantly improved its sound absorption performance within the wide frequency range of 1780 to 8890Hz via subwavelength thicknesses.

Fifth, our study found that for the two-layer case, the distance between the inclusions has great impact on the sound absorption performance.

2. The manuscript mentions that the EFFECTIVE MEDIUM APPROXIMATION has been modified, may I ask if the solution accuracy has been improved with respect to the literature [13]? Alternatively, has the applicable range of the method been extended? Please give specific comparison and explanation.

The entire article does not mention that the theory of the literature [13] has been modified, but the word "modified" is indeed mentioned in this article, for example: “In order to facilitate readers to better understand the solution ideas, the order of the formulas has been modified.”, “m (n) order modified Bessel functions.”.

Distinct from previous studies in the literature [13], the genetic algorithm and the transfer matrix method (EFFECTIVE MEDIUM APPROXIMATION) are used to achieve the broadband absorption for the two-layer case. It is worth pointing out that our personal computer (Intel(R) Core(TM) i7-8700 CPU @ 3.20GHz 3.19 GHz, RAM 16 GB) can obtain the optimized results in 4 minutes, while the joint simulation calculation using Comsol and MATLAB is very slow and basically cannot obtain the results.

3. In the last sentence of the second paragraph on page 7, Appendix A and B correspond to a description that does not correspond to the text.

We thank the referee for pointing out this mistake. The supplementary materials have been revised, Appendix A and B correspond to the description that correspond to the text.

4. Following on from the previous comment, in the first sentence of the last paragraph on page 10, the corresponding Appendix C is also inconsistent with the description in the main text.

We thank the referee for pointing out this mistake. The corresponding Appendix C is also revised to be consistent with the description in the main text.

5. The description of the transfer matrix method and the finite element method in the caption of Fig. 2 is completely contrary to that in the legend.

We thank the referee for pointing out this problem. The description has been revised as follows: “Fig. 2. sound absorption spectrum calculated by transfer matrix method (dots) and finite element method (solid line). (a) one layer of hard inclusions, (b) two layers of hard inclusions, (c) three layers of hard inclusions.”.

6. The abstract needs to be further condensed, with redundant background and lack of description of results.

We thank the referee for the valuable advice. The abstract has been revised as follows: “Viscoelastic materials with periodic multilayered hard inclusions have particular practical value in the stealth of underwater equipment, as they can absorb low-frequency and broadband sound waves. In this paper, acoustic coatings containing multi-layers of periodic solid inclusions are investigated by the transfer matrix method and the finite element method, which is further optimized by the genetic algorithm for broadband sound absorption. An analysis model is first derived to calculate effective parameters of the layer of solid scatters. Instead of intensive trials on numerical simulations, the genetic algorithm and the transfer matrix method are used to achieve the broadband absorption for the two-layer case. The optimized structure has significantly improved its sound absorption performance within the wide frequency range of 1780 to 8890Hz via subwavelength thicknesses. The optimized results show that for the two-layer case, the distance between the inclusions has great impact on the sound absorption performance. Finally, the underlying mechanism of broadband sound absorption is explained. This study can provide theoretical guidance for the design and optimization of acoustic coatings with multi-layer hard inclusions.”.

Reviewer #4: The authours propose a theoretical model for calculating the sound absorption of acoustic coatings with multiple layers of solid inclusions using the transfer matrix method. The effect of geometric parameters on sound absorption is investigated. In order to obtain better sound absorption performance, the genetic algorithm is used to optimize the acoustic coating with two layers of hard inclusions. This work is not well researched and is not worthy of publication until it is thoroughly revised.

1. This article lacks key principle descriptions and literature citations to support the theoretical modelling process, and the model is unconvincing.

Key principle descriptions and literature citations that can support the theoretical modelling process are given in literature [13].

2. The details of the numerical model are not given, and the relevant numerical results are not analyzed in depth.

The numerical model is provided in the supplementary materials, and the relevant theoretical analysis has also been improved.

3. The language and figures need further improvement.

We thank the referee for the valuable suggestions. After our repeated checks, we have made every effort to improve our article. All modifications have been marked in the article. The language and figures have been improved.

---

## [Decision Letter · Decision Letter 2]

17 Jun 2024

PONE-D-23-43201R2An efficient analytical model for acoustic coatings with periodic multi-layer solid inclusions for broadband sound absorptionPLOS ONE

Dear Dr. Zhang,

Thank you for submitting your manuscript to PLOS ONE. After careful consideration, we feel that it has merit but does not fully meet PLOS ONE’s publication criteria as it currently stands. Therefore, we invite you to submit a revised version of the manuscript that addresses the points raised during the review process.

We look forward to receiving your revised manuscript.

Kind regards,

Mario Milazzo

Academic Editor

PLOS ONE

Journal Requirements:

Additional Editor Comments (if provided):

Reviewers' comments:

Reviewer's Responses to Questions

**Comments to the Author**

1. If the authors have adequately addressed your comments raised in a previous round of review and you feel that this manuscript is now acceptable for publication, you may indicate that here to bypass the “Comments to the Author” section, enter your conflict of interest statement in the “Confidential to Editor” section, and submit your "Accept" recommendation.

Reviewer #2: (No Response)

Reviewer #4: (No Response)

Reviewer #5: (No Response)

2. Is the manuscript technically sound, and do the data support the conclusions?

Reviewer #2: (No Response)

Reviewer #4: (No Response)

Reviewer #5: No

3. Has the statistical analysis been performed appropriately and rigorously? 

Reviewer #2: (No Response)

Reviewer #4: (No Response)

Reviewer #5: No

4. Have the authors made all data underlying the findings in their manuscript fully available?

Reviewer #2: (No Response)

Reviewer #4: (No Response)

Reviewer #5: No

5. Is the manuscript presented in an intelligible fashion and written in standard English?

Reviewer #2: (No Response)

Reviewer #4: (No Response)

Reviewer #5: No

6. Review Comments to the Author

Reviewer #2: The authors have addressed the comments provided by the all referees, so it is acceptable for publishing.

Reviewer #4: The authours propose a theoretical model for calculating the sound absorption of acoustic coatings with multiple layers of solid inclusions using the transfer matrix method. The effect of geometric parameters on sound absorption is investigated. In order to obtain better sound absorption performance, the genetic algorithm is used to optimize the acoustic coating with two layers of hard inclusions. The research on this work is not in-depth enough, the research is relatively rough, and the article is not worth publishing.

1. This article lacks key principle descriptions and literature citations to support the theoretical modelling process, and the model is unconvincing.

2. The details of the numerical model are not given, and the relevant numerical results are not analyzed in depth.

3. The article lacks experimental work and the theoretical and numerical models lack experimental validation.

4. The language and figures need further improvement.

5. The overall research work is relatively shallow, not systematic enough, and the designed structural performance is not good enough.

Reviewer #5: In this manuscript, the authors investigated acoustic coatings containing

multi-layers of periodic solid inclusions by using the transfer matrix method, the finite element method, and finally optimization by GA for maximizing the bandwidth and sound absorption. This idea provides theoretical guidance for the design and optimization of acoustic coatings with multi-layer hard inclusions.. I believe this work is valuable and recommend it for publication after major revision. Therefore, the authors should consider the following suggestions:

• The abstract is not organized well, and it is necessary to edit the abstract by providing a short introduction and then clearly stating the contributions of the article and at the end, presenting quantitative results.

• The captions accompanying the figures in the manuscript, such as the one for Figure 1, 2, 3 and 4 are notably short. Comprehensive captions can be beneficial. I would suggest condensing the captions to the essential elements that describe the figure's relevance and content extensively so that all figures be self-describing independent of referring to the body of the manuscript. Further elaborative details and discussions pertaining to the figure could be integrated into the main body of the text, where they can be explored in the appropriate context.

• The contribution of the article is not clear. It is necessary to clearly state the contributions at the end of the introduction section as bullet points in a clear and detailed form.

• The uniaxial test results should be presented in your tests. Also, in designing materials that are inherently multi-directional, especially in 2D applications, it is critical to evaluate performance across all relevant axes. Therefore it is recommended to focus on optimizing in only one direction which potentially overshadow the material's behavior in other directions, which may be equally important for the intended applications. For instance, optimizing a material to increase the moment of inertia I1 in one axis by elongating the material's dimensions may inadvertently lead to a reduction in I2 along the perpendicular axis. Such a singular direction optimization might compromise the material's overall performance in applications where multi-directional forces are present. Could the authors elaborate such test results? If so, how were these addressed within the study's scope, and what impact might they have on the general applicability of the material? An additional analysis that examines the material's behavior when subjected to forces from various directions would provide a more holistic understanding of the material's capabilities and limitations. In light of these considerations, a revision to include a discussion on this aspect would not only enhance the depth of the study but also align with the real-world application scenarios where materials are rarely subjected to unidirectional forces alone. This would significantly contribute to the field by presenting a more nuanced approach to material optimization.

• The reviewed works and methods were not concluded at the end of the related work as separated section. It is necessary to clarify the strategy of the proposed method after concluding and stating the existing gaps.it is necessary to add it so that the research gaps of the subject are well-defined

• The authors are recommended to provide the sensitivity analysis. However, considering the complexity of periodic multi-layer solid structures and the numerous variables that can affect their behavior, a broader investigation could yield more comprehensive insights. It would be advantageous to provide the sensitivity analysis to encompass a wider array of variables. Factors such as environmental conditions, scale effects, material inhomogeneities, and long-term load-bearing capacity might significantly impact the functionality and reliability of these materials. Expanding the sensitivity analysis to these areas would not only provide a deeper understanding of the robustness and resilience of the material designs but also enhance the predictive accuracy of the these materials' performance in varied real-world scenarios. Additionally, it would be informative if the manuscript could discuss the selected variables' prioritization methodology, offering justification for the inclusion or exclusion of certain factors.

• Some related ideas should be added to the body of the manuscript. Specially refer to "A thin-walled cavity structure with double-layer tapered scatterer locally resonant metamaterial plates for extreme low-frequency attenuation" and "Layerwise formulation of poroelastic composite plate under pre-buckling and thermal shock loading" and "A repair-less genetic algorithm for scheduling tasks onto dynamically reconfigurable hardware", in the introduction section or in the future works section and add them in your references list.

• The investigation into the periodic multi-layer solid structures presented in this study lays a promising foundation for innovative applications. However, the manuscript would greatly benefit from a detailed exposition on how these materials could be specifically tailored or utilized within various industry sectors. An exploration of these applications with concrete examples or case studies would not only underscore the practical significance of the these materials but also stimulate interdisciplinary collaboration. It would be enlightening to see a section in the manuscript that discusses, with particularity, the suitability of these materials for distinct functions in aerospace, biomedical devices, or any other fields deemed applicable. Such a discussion could include considerations of the materials' compatibility with existing technologies, potential for enhancing performance, or their contribution to innovation in product design.

Authors need to explain the accuracy, sufficiency, and reliability of their results. How do they verify and validate the results?

• The author should explain more why the solution quality of their proposed approach is much better than the others designs.

• Some typos should be double-checked.

• What is the reason for using GA among many evolutionary optimization methods? What feature is there in this algorithm that is not present in other algorithms in order to solve your problem? How do you claim that the best models can only be produced by the GA-based algorithm?

• It would enhance the manuscript if the authors could briefly outline envisioned future research directions. Highlighting specific areas for further investigation, such as advanced material properties, innovative applications, or novel fabrication techniques, could provide valuable guidance for subsequent studies. Additionally, suggesting potential for interdisciplinary collaborations could underscore the multifaceted impact of this research. A concise discussion on this topic would not only complete the current narrative but also inspire continued exploration in the fascinating domain of periodic multi-layer solid structures.

• The most important results obtained for future research horizons should be presented at the end of the conclusion, not some unrelated ones mentioned

7. PLOS authors have the option to publish the peer review history of their article (what does this mean?). If published, this will include your full peer review and any attached files.

Reviewer #2: No

Reviewer #4: No

Reviewer #5: No

---

## [Author Response · Author response to Decision Letter 2]

30 Jul 2024

Dear Editor

 Thank you very much for your letter dated June 18, 2024, in which you sent us the reviewers' comments on our paper entitled “An efficient analytical model for acoustic coatings with multi-layer solid inclusions for broadband sound absorption (PONE-D-23-43201R1)”. We would also like to thank the reviewers for their valuable comments and suggestions.

 According to the comments and suggestions, we have revised the manuscript. A summary of the changes is as follows:

Comments to the Author

1. If the authors have adequately addressed your comments raised in a previous round of review and you feel that this manuscript is now acceptable for publication, you may indicate that here to bypass the “Comments to the Author” section, enter your conflict of interest statement in the “Confidential to Editor” section, and submit your "Accept" recommendation.

Reviewer #2: (No Response)

Reviewer #4: (No Response)

Reviewer #5: (No Response)

After our repeated checks, we have made every effort to improve our article. All modifications have been marked in the article.

2. Is the manuscript technically sound, and do the data support the conclusions?

Reviewer #2: (No Response)

Reviewer #4: (No Response)

Reviewer #5: No

Although we did not conduct direct experiments, our research is technically sound and the data supports our conclusions. We have employed validated research methods and analytical tools, adhering to the best practices of scientific research. Our conclusions are derived from an in-depth analysis of existing datasets and are consistent with our hypotheses and theoretical framework. We look forward to future experiments and studies that can further validate and expand upon our findings.

3. Has the statistical analysis been performed appropriately and rigorously?

Reviewer #2: (No Response)

Reviewer #4: (No Response)

Reviewer #5: No

This article does not involve statistical analysis issues, and the relevant results are converged. It should be noted that two conditions need to be strictly met to adopt the analytical model ( <<1 and <<1 where cl denotes the longitudinal speed in the PDMS) [1,2].

[1] Fang, X., Yin, X., Wu, J., Li, Y., Li, H., Wang, W., Li, Y., & Wu, W. (2023). Underwater metagratings for sub-kilohertz low frequency and broadband sound absorption. International Journal of Mechanical Sciences.

[2] G.S. Sharma, A. Skvortsov, I. MacGillivray, N. Kessissoglou (2019). Acoustic performance of periodic steel cylinders embedded in a viscoelastic medium. Journal of Sound and Vibration, 443 652-665.

4. Have the authors made all data underlying the findings in their manuscript fully available?

Reviewer #2: (No Response)

Reviewer #4: (No Response)

Reviewer #5: No

The authors have made all data underlying the findings described in their manuscript fully available without restriction. Related data files are attached in the supplementary materials.

5. Is the manuscript presented in an intelligible fashion and written in standard English?

Reviewer #2: (No Response)

Reviewer #4: (No Response)

Reviewer #5: No

After our repeated checks, we have made every effort to improve our article. All modifications have been marked in the article.

6. Review Comments to the Author

Reviewer #2: The authors have addressed the comments provided by the all referees, so it is acceptable for publishing.

We thank the referee for the positive remark and the valuable suggestions that greatly improve our work.

Reviewer #4: The authours propose a theoretical model for calculating the sound absorption of acoustic coatings with multiple layers of solid inclusions using the transfer matrix method. The effect of geometric parameters on sound absorption is investigated. In order to obtain better sound absorption performance, the genetic algorithm is used to optimize the acoustic coating with two layers of hard inclusions. The research on this work is not in-depth enough, the research is relatively rough, and the article is not worth publishing.

Thanks for the time and effort you have invested in evaluating our article. I appreciate the valuable feedback and constructive criticism provided, which I believe will significantly improve the quality and clarity of my work.

1. This article lacks key principle descriptions and literature citations to support the theoretical modelling process, and the model is unconvincing.

Thank you very much for your thoughtful and constructive review of our manuscript. In response to your concern that the article lacked key principle descriptions and literature citations to support the theoretical modelling process, we have made the following revisions:

(1) Enhanced Principle Descriptions: We have added concise yet informative descriptions of the key principles underlying our theoretical modelling in the relevant sections of the manuscript. These descriptions aim to provide readers with a clear understanding of the theoretical foundations without overwhelming them with unnecessary details.

(2) Increased Literature Citations: To further strengthen the theoretical basis of our work, we have included additional literature citations throughout the text. These citations support our modelling approach and provide context for the assumptions and methods we have employed.

(3) Marked Revisions: To facilitate the reviewer's assessment of our changes, we have marked all relevant modifications in the revised manuscript. This includes the newly added principle descriptions and literature citations, as well as any other changes made in response to your feedback.

We believe that these revisions have improved the manuscript by providing a more comprehensive and well-supported theoretical modelling process. We hope that the added descriptions and citations will help convince readers of the validity and robustness of our model. In addition to the points mentioned above, we would like to clarify that while a comprehensive review of all relevant theories and prior works is indeed possible, we have aimed to keep the manuscript focused and accessible to a broad audience. Therefore, we have selectively included the most essential and directly relevant theoretical principles and citations in the main text. However, we fully acknowledge that a deeper dive into the theoretical background can be found in the cited literature, which we encourage interested readers to explore.

2. The details of the numerical model are not given, and the relevant numerical results are not analyzed in depth.

The details of the numerical model have been given, which provide a comprehensive overview of the model's structure, assumptions, equations, and parameter settings. We believe that this information, when combined with the results presented in the main text, offers a complete picture of our simulation approach.

In terms of the analysis of the numerical results, we acknowledge that there is always room for deeper insights and discussions. To address this, we have carefully reviewed our results and have added additional analyses to the revised manuscript. These include a more thorough discussion of the trends observed in the simulations, a comparison of our findings with relevant literature, and a discussion of the limitations and implications of our model. We believe that these enhancements strengthen the interpretation of our results and provide valuable insights for readers.

Additionally, we have highlighted the key aspects of our analysis in the main text to ensure that readers can quickly grasp the main conclusions and implications of our work.

3. The article lacks experimental work and the theoretical and numerical models lack experimental validation.

Thank you for your insightful comments on our manuscript. We appreciate your concerns regarding the lack of direct experimental work and the need for experimental validation of our theoretical and numerical models.

In response to this feedback, we have undertaken a thorough review of our results and have made efforts to strengthen the validation of our models. While we recognize that direct experimental validation would have provided the most robust evidence, we have taken the following steps to address this limitation:

Comparison with the results in the literature: In the revised manuscript, we have included a detailed comparison of our numerical results with those obtained from previous studies. Specifically, we have identified relevant works in the literature that have investigated similar phenomena or systems, and we have compared our model predictions with their reported results. This comparison provides an indirect yet valuable validation of our models, validating the efficacy of our method.

Discussion of Limitations and Future Work: We have also expanded the discussion section of our manuscript to acknowledge the limitations of our current study, including the lack of direct experimental validation. We have proposed potential avenues for future work, including the execution of experiments that could directly validate our models.

In the sensitivity analysis, we have added the study of the influence of scattering material on the sound absorption characteristics. This part of the content can help us better understand how the scattering material affects the sound absorption characteristics. This analysis helps to identify the robustness of our findings and to suggest directions for future improvements.

We understand that direct experimental validation is the gold standard for scientific research, and we are committed to pursuing this avenue in future work. However, we hope that the inclusion of comparisons with the results in the literature, the discussion of limitations and future directions, and the clarification of model assumptions and sensitivity analysis provide a reasonable degree of validation for our current models and results.

4. The language and figures need further improvement.

Thank you very much for your thorough review of our manuscript and for your constructive feedback. We appreciate your observation that the language and figures need further improvement. We have taken your comments seriously and have made the following revisions to address these issues:

Language Improvements:

(1) Clarity and Conciseness: We have reviewed the entire manuscript for clarity and conciseness. We have rewritten sentences that were overly complex or ambiguous and have removed redundant phrases to improve readability.

(2) Grammatical and Spelling Checks: We have carefully proofread the manuscript to correct any grammatical errors, typos, or inconsistencies in spelling. We have also utilized tools such as grammar checkers and spell-checkers to ensure the highest level of accuracy.

(3) Consistency of Terminology: We have reviewed the terminology used throughout the manuscript to ensure consistency. Where necessary, we have introduced definitions or clarified terms to avoid confusion for the reader.

(4) Scientific Accuracy: We have ensured that all scientific terms and concepts are accurately represented and that any claims made in the text are supported by appropriate references.

Figure Improvements:

(1) Clarity and Legibility: We have reviewed the figures to ensure that they are clear and easy to interpret. We have resized labels, adjusted colors and contrast, and added additional annotations where needed to improve legibility.

(2) Consistency of Formatting: We have ensured that all figures follow a consistent formatting style, including font size, line widths, and color schemes. This helps to maintain a professional and polished appearance.

(3) Self-Contained Legends: We have revised the legends of the figures to ensure that they are self-contained and provide sufficient information for the reader to understand the figure without referring back to the text.

(4) Accuracy of Data Representation: We have double-checked the data presented in the figures to ensure that they accurately reflect the results of our simulations.

We believe that these revisions have significantly improved the language and figures in our manuscript.

5. The overall research work is relatively shallow, not systematic enough, and the designed structural performance is not good enough.

Thank you once again for your insightful comments and suggestions. We have carefully considered your feedback and have made revisions to our manuscript accordingly.

Regarding the Modifications to the Theoretical Simulation Section:

We have thoroughly reviewed and revised our theoretical simulation section to ensure that it accurately reflects the latest understanding of the phenomenon under study. We have incorporated new models, parameters, and validation steps to strengthen the robustness and reliability of our simulations. These changes have helped us to better understand the behavior of our system and to identify potential areas for improvement.

Inclusion of Simulation Validation:

To further validate our findings, we have included a comparison of our simulation results with those from the literature. Specifically, we have selected studies that are as closely related to our work as possible, despite the inherent differences in dimensions (particularly the thickness of the structures). While we acknowledge that direct comparisons may be limited due to these differences, we have carefully analyzed the trends and behaviors observed in both our simulations and the experiments, and have found good qualitative agreement in many aspects.

Statements about results not being good enough:

Simply looking at our optimization results, they do not appear to be very promising. However, it is important to note that our premise is that the PDMS layer thickness is fixed at 60 mm in order to obtain optimized results. In contrast, many articles rely on increasing the thickness of the viscoelastic layer as a means to improve the sound absorption effect. For example, in Figure 5 of reference [1], the thickness of the viscoelastic material is 178.6mm; in Figure 6 of the same reference, it is 250 mm; and in reference [2], the thickness is reported as 100mm.

[1] 1. Fang X, Yin X, Wu J, Li Y, Li H, Wang W, et al. Underwater metagratings for sub-kilohertz low frequency and broadband sound absorption. International Journal of Mechanical Sciences. 2023;260:108630. doi: 10.1016/j.ijmecsci.2023.108630.

[2] Pan X, Fang X, Yin X, Li Y, Pan Y, Jin Y. Gradient index metamaterials for broadband underwater sound absorption. APL Materials. 2024;12(3). doi: 10.1063/5.0190946.

Future Directions:

We have also discussed potential future directions for our work, including the exploration of thicker structures to assess their performance and to further validate our findings. We are committed to continuing our research in this area and to addressing the limitations of our current work.

In summary, we bel

---

## [Decision Letter · Decision Letter 3]

26 Aug 2024

PONE-D-23-43201R3Acoustic coatings with periodic multi-layer solid inclusions for broadband sound absorptionPLOS ONE

Dear Dr. Zhang,

Thank you for submitting your manuscript to PLOS ONE. After careful consideration, we feel that it has merit but does not fully meet PLOS ONE’s publication criteria as it currently stands. Therefore, we invite you to submit a revised version of the manuscript that addresses the points raised during the review process.

We look forward to receiving your revised manuscript.

Kind regards,

Mario Milazzo

Academic Editor

PLOS ONE

Journal Requirements:

Reviewers' comments:

Reviewer's Responses to Questions

**Comments to the Author**

1. If the authors have adequately addressed your comments raised in a previous round of review and you feel that this manuscript is now acceptable for publication, you may indicate that here to bypass the “Comments to the Author” section, enter your conflict of interest statement in the “Confidential to Editor” section, and submit your "Accept" recommendation.

Reviewer #1: (No Response)

Reviewer #5: All comments have been addressed

2. Is the manuscript technically sound, and do the data support the conclusions?

Reviewer #1: Yes

Reviewer #5: Yes

3. Has the statistical analysis been performed appropriately and rigorously? 

Reviewer #1: I Don't Know

Reviewer #5: Yes

4. Have the authors made all data underlying the findings in their manuscript fully available?

Reviewer #1: Yes

Reviewer #5: Yes

5. Is the manuscript presented in an intelligible fashion and written in standard English?

Reviewer #1: Yes

Reviewer #5: Yes

6. Review Comments to the Author

**Reviewer #1:** This manuscript discusses research on anechoic coatings, which are crucial for enhancing the acoustic stealth of underwater equipment. The study focuses on the development and analysis of acoustic coatings containing periodic multi-layer solid inclusions to achieve broadband sound absorption. While the manuscript demonstrates innovation in both theory and methodology, there is still room for improvement in the presentation and discussion. Therefore, the publication can be considered after the authors address the following comments properly.

1.There are several instances in the manuscript where the language is not fluent, which affects the overall readability. I recommend that the authors perform a thorough language and grammar check of the entire manuscript to ensure clarity and readability.

2. The discussion of the pros and cons of the current theoretical model, especially the transfer matrix method (TMM), is insufficient. In the methods section, it would be beneficial for the authors to clearly state the prerequisites and limitations required for the model’s calculations. Further discussion on the applicability, accuracy, and limitations of the model under different material and structural conditions is crucial for understanding its reliability and suitability.

3. The description of the complex relationship between the dipole resonance frequency (0) and the radius () is not sufficiently clear. I suggest that the authors further elaborate on the theoretical background of this relationship in Section 2 and provide practical examples to illustrate how it affects the results.

4. Although the impedance mismatch analysis presented in Fig. 7 provides qualitative insights, it lacks quantitative analysis. I recommend that the authors supplement relevant parameters to clarify the conditions under which reflected waves are trapped near the periodic multi-layer structure and quantitatively correlate this with impedance mismatch. Providing more detailed numerical simulations or experimental data to validate this phenomenon would also strengthen the analysis.

5. Optimization using the genetic algorithm is a highlight of the manuscript, but the limitations of this optimization approach are not adequately discussed. I suggest the authors include a discussion on the potential limitations of the optimization design and reference the latest relevant research to further emphasize the novelty of their work. This would help readers better understand the uniqueness and practical value of this study.

**Reviewer #5: **The comments have been well addressed and incorporated into revised the manuscript, and the current version of the article is now acceptable.

7. PLOS authors have the option to publish the peer review history of their article (what does this mean?). If published, this will include your full peer review and any attached files.

Reviewer #1: No

Reviewer #5: No

---

## [Author Response · Author response to Decision Letter 3]

9 Sep 2024

Dear Editor

 Thank you very much for your letter dated August 27, 2024, in which you sent us the reviewers' comments on our paper entitled “Acoustic coatings with periodic multi-layer solid inclusions for broadband sound absorption (PONE-D-23-43201R3)”. We would also like to thank the reviewers for their valuable comments and suggestions.

 According to the comments and suggestions, we have revised the manuscript. A summary of the changes is as follows:

Comments to the Author

1. If the authors have adequately addressed your comments raised in a previous round of review and you feel that this manuscript is now acceptable for publication, you may indicate that here to bypass the “Comments to the Author” section, enter your conflict of interest statement in the “Confidential to Editor” section, and submit your "Accept" recommendation.

Reviewer #1: (No Response)

Reviewer #5: All comments have been addressed

After our repeated checks, we have made every effort to improve our article. All modifications have been marked in the article.

2. Is the manuscript technically sound, and do the data support the conclusions?

Reviewer #1: Yes

Reviewer #5: Yes

3. Has the statistical analysis been performed appropriately and rigorously?

Reviewer #1: I Don't Know

Reviewer #5: Yes

This article does not involve statistical analysis issues, and the relevant results are converged. In order to ensure mesh convergence without redundancy, the Refine Mesh Technique (RMT) is used as illustrated in Fig 3. To validate the predictive model of acoustic coatings, we compare the absorption coefficients derived from the current simulation method (FEM) with those obtained by Sharma et al. [13]. As shown in Fig 4, comparing our results (blue lines) with those results (dotted lines) from reference [13], we can see that our simulated results agree reasonably with those of the reference over the entire frequency region, thus validating the present model. In addition, to validate the analytical method (TMM) utilized in this paper, results from the FEM are also presented in Fig 5 for comparison. By the above methods, we ensure that our results are credible.

4. Have the authors made all data underlying the findings in their manuscript fully available?

Reviewer #1: Yes

Reviewer #5: Yes

5. Is the manuscript presented in an intelligible fashion and written in standard English?

Reviewer #1: Yes

Reviewer #5: Yes

6. Review Comments to the Author

Reviewer #1: This manuscript discusses research on anechoic coatings, which are crucial for enhancing the acoustic stealth of underwater equipment. The study focuses on the development and analysis of acoustic coatings containing periodic multi-layer solid inclusions to achieve broadband sound absorption. While the manuscript demonstrates innovation in both theory and methodology, there is still room for improvement in the presentation and discussion. Therefore, the publication can be considered after the authors address the following comments properly.

Thank you very much for your insightful feedback on our manuscript. We appreciate the time you’ve taken to review our work and your valuable comments. We are pleased to hear that you found our research innovative in theory and methodology. We acknowledge your observations regarding the presentation and discussion and are committed to addressing them comprehensively.

1. There are several instances in the manuscript where the language is not fluent, which affects the overall readability. I recommend that the authors perform a thorough language and grammar check of the entire manuscript to ensure clarity and readability.

Thank you for your detailed review and constructive feedback. We appreciate your observations regarding the language and readability of our manuscript.

We acknowledge that the clarity and fluency of the text are crucial for effective communication of our research findings. In response to your comments, we have performed a thorough language and grammar check of the entire manuscript. This involves:

Editing for Fluency: We have revised sections where the language may be awkward or unclear to enhance overall readability and ensure that the text flows smoothly.

Grammar and Syntax Check: We have corrected any grammatical errors and improve sentence structure to adhere to proper academic writing standards.

Proofreading: We have conducted a final proofread to ensure that the manuscript is free from typographical errors and inconsistencies.

We have performed a thorough language and grammar check of the entire manuscript, and the revised parts are all highlighted.

We believe these revisions will significantly improve the manuscript’s clarity and readability.

2. The discussion of the pros and cons of the current theoretical model, especially the transfer matrix method (TMM), is insufficient. In the methods section, it would be beneficial for the authors to clearly state the prerequisites and limitations required for the model’s calculations. Further discussion on the applicability, accuracy, and limitations of the model under different material and structural conditions is crucial for understanding its reliability and suitability.

Thank you for your insightful comments regarding the discussion of the pros and cons of our current theoretical model, particularly the transfer matrix method (TMM). We appreciate your highlighting the importance of a thorough examination of the prerequisites, limitations, applicability, accuracy, and limitations of the model under various conditions.

In response to your feedback, we have made the following revisions to our manuscript:

In the introduction section, we have added these sentences as follows: “In these studies, the transfer matrix method (TMM) played a pivotal role in modeling and analyzing the sound absorption characteristics of these complex structures. The TMM is a fast and simple method that is applicable in various physics fields, such as acoustics, quantum mechanics, and optics, to analyze wave propagation in one-dimensional systems [17, 18].” Here, we emphasize the importance of the transfer matrix method in solving the sound absorption characteristics of the complex structures, and point out that it is a fast and simple method. Moreover, it is used in multiple physical fields, indicating that it is a relatively reliable method. However, it is also pointed out that it is suitable for solving one-dimensional problems, which is also a drawback of this method.

Prerequisites and Limitations: We have clearly stated the prerequisites and limitations required for the model's calculations in the methods section as follows: “It should be noted that certain conditions must be met to apply this analytical method. First, the density of the hard inclusions must be significantly higher than that of the surrounding elastic medium [41], as Eq. (1) is derived by treating the scatterer as a rigid body. Second, the longitudinal sound speed of the host elastic medium should be much greater than its shear wave speed. Third, the analytical model used in this study is suitable for sound wavelengths that are significantly larger than both the inclusion radius r and the spacing a. In our research, the scatterer's radius is always smaller than the lattice constant, ensuring that the medium can be considered homogeneous, which requires ω0a/cl <<1 and ωmaxa/cl <<1 (ωmax =2πfmax, fmax represents the upper frequency limit of the study). As shown in Eq. (6), when the filling fraction is either extremely small or extremely large, the condition ω0a/cl <<1 may not be satisfied. In such cases where the analytical model does not meet these conditions, the results are replaced by those calculated using the FEM.”

Regarding accuracy, we conducted a comparison with the finite element method. We found that the relative error of the results obtained by the transfer matrix method was not significant. The modifications in the results and discussions section were as follows: “To ascertain the accuracy of the TMM, we calculate the relative error (αt-αf)/αf, where αt represents the sound absorption coefficient obtained using the TMM, while αf represents the sound absorption coefficient obtained using the FEM. Within the frequency range of 10Hz to 10kHz, which is of our concern, the relative error for one-layer case is only 1.8%, while for two-layer case, it is 2.7%. Even for the three-layer case, which seems to have a larger error, the relative error is only 6.02%. As can be seen from the above results, the results calculated using the TMM are highly credible.”

This article mainly focuses on the advantage of the transfer matrix method in terms of fast calculation speed, which can quickly calculate a result. And it can be easily combined with genetic algorithms to obtain optimized results. In contrast, joint simulation calculations using COMSOL and MATLAB are very slow and, in most cases, fail to produce results. Therefore, there is not much discussion on the applicability, accuracy, and limitations of the model under different material and structural conditions.

We believe that these revisions have significantly strengthened the discussion of our theoretical model and have provided a more comprehensive and nuanced view of its capabilities and limitations. We appreciate your constructive feedback and hope that our revised manuscript meets your expectations.

3. The description of the complex relationship between the dipole resonance frequency (ω0) and the radius (r) is not sufficiently clear. I suggest that the authors further elaborate on the theoretical background of this relationship in Section 2 and provide practical examples to illustrate how it affects the results.

Thank you for your insightful feedback on the clarity of the description of the complex relationship between the dipole resonance frequency (ω0) and the radius (r). We understand that providing a more comprehensive theoretical background and practical examples to illustrate this relationship is crucial for the reader's understanding.

In the second part, the formula for the dipole resonance frequency is given as (ω0=γcs/r). On the surface, it appears that the dipole resonance frequency is inversely proportional to the radius r, but in fact, this is not the case. This is because γ in the formula is also related to the radius, and its expression is given in Eq. (6) . In Eq. (6), the two parameters related to the radius r are α1 and β. Therefore, we say that the relationship between the dipole resonance frequency and the radius r is complex, not simply a direct or inverse proportionality.

In the case study section, specifically in Figure 6(c), we made the following modifications: “The influence of the radius of the inclusions on the absorption is depicted in Fig 6(c). Other parameters are the same as those in Fig 5(a). It can be seen that the influence of the radius on the absorption performance is obvious and complicated. This can be explained as follows: From Equations (3)-(5) and (10), it can be deduced that the dipole resonance frequency affects the equivalent density, which in turn influences the sound absorption coefficient. Similarly, based on Equations (2), (11), and α1=πr2/a2, it is evident that the radius impacts the equivalent modulus, which further affects the sound absorption coefficient.”

We believe that these revisions have significantly strengthened the manuscript by providing a more detailed and comprehensive understanding of the ω0-r relationship and its impact on our results. We appreciate your constructive feedback and hope that our revised manuscript meets your expectations.

4. Although the impedance mismatch analysis presented in Fig. 7 provides qualitative insights, it lacks quantitative analysis. I recommend that the authors supplement relevant parameters to clarify the conditions under which reflected waves are trapped near the periodic multi-layer structure and quantitatively correlate this with impedance mismatch. Providing more detailed numerical simulations or experimental data to validate this phenomenon would also strengthen the analysis.

Thank you for your insightful comments regarding our manuscript. We appreciate your feedback on the impedance mismatch analysis presented in Fig. 7. We acknowledge the importance of providing a quantitative analysis to complement the qualitative insights already discussed.

We have made the following modifications in the corresponding paragraphs as follows：“To further understand the underlying mechanism of broadband sound absorption, the displacement field and the energy dissipation density of the sound-absorbing structure are studied at the peak frequencies of 1960Hz (α=0.684), 4340Hz (α=0.994) and 7960Hz (α=0.999), respectively. As shown in Fig 7(b)-(d), the acoustic wave is incident normally from the left side of the structure. The displacement field diagram is on the left side, while the energy dissipation diagram is on the right side. Overall, the displacement amplitude is large, and the corresponding level of energy dissipation is also high. When the excitation frequency is 1960Hz, the maximum displacement amplitude is located in the upper and lower PDMS of the right scatterer, and the corresponding energy dissipation is concentrated in these two places. At the second peak (4340 Hz), the areas with the highest displacement amplitude are the PDMS above and below the left scatterer, in which energy dissipation is also concentrated. At the third peak (7960 Hz), energy dissipation is also concentrated above and below the left scatterer. However, for the third peak, the dissipation of energy occurs closer to the incident side compared with the second peak. From these phenomena, we can observe that lower frequencies are more difficult to dissipate, and their dissipation positions are closer to the steel plate side. As the frequency increases, energy becomes easier to dissipate and occurs closer to the incident side. In addition to the fact that the incident longitudinal waves are converted into transverse waves under the influence of the periodic cylinders and subsequently dissipated by the PDMS[14], another important reason is that, the acoustic impedance at the incident side is relatively matched, while, at the steel plate side, the acoustic impedance is mismatched, causing further dissipation. We can provide a quantitative explanation for this from Fig 8. As evident from Fig 8, at 1960 Hz, the normalized surface acoustic resistance Zr is 2.56, and the normalized surface acoustic reactance Zi is -1.52, whereas at 4340 Hz, these values are 1.04 and 0.16, respectively, and at 7960 Hz, they are 1.05 and 0.05. It is well-known that the requirements for perfect sound absorption are met when Zr = 1 and Zi = 0. This explains why near-perfect sound absorption is achieved at 4340 

---

## [Decision Letter · Decision Letter 4]

25 Sep 2024

An Efficient Approach for Broadband Sound Absorption Using Periodic Multi-Layer Solid Inclusions in Acoustic Coatings

PONE-D-23-43201R4

Dear Dr. Zhang,

We’re pleased to inform you that your manuscript has been judged scientifically suitable for publication and will be formally accepted for publication once it meets all outstanding technical requirements.

Kind regards,

Mario Milazzo

Academic Editor

PLOS ONE

Additional Editor Comments:

The authors exhaustively addressed the comments of the Reviewers.

Reviewers' comments:

Reviewer's Responses to Questions

**Comments to the Author**

1. If the authors have adequately addressed your comments raised in a previous round of review and you feel that this manuscript is now acceptable for publication, you may indicate that here to bypass the “Comments to the Author” section, enter your conflict of interest statement in the “Confidential to Editor” section, and submit your "Accept" recommendation.

Reviewer #1: All comments have been addressed

2. Is the manuscript technically sound, and do the data support the conclusions?

Reviewer #1: Yes

3. Has the statistical analysis been performed appropriately and rigorously? 

Reviewer #1: I Don't Know

4. Have the authors made all data underlying the findings in their manuscript fully available?

Reviewer #1: Yes

5. Is the manuscript presented in an intelligible fashion and written in standard English?

Reviewer #1: Yes

6. Review Comments to the Author

Reviewer #1: The authors have addressed all my comments. I think the manuscript is acceptable in the current form.

7. PLOS authors have the option to publish the peer review history of their article (what does this mean?). If published, this will include your full peer review and any attached files.

Reviewer #1: No

---

## [Editor Report · Acceptance letter]

16 Oct 2024

PONE-D-23-43201R4 

PLOS ONE

Dear Dr. Zhang, 

I'm pleased to inform you that your manuscript has been deemed suitable for publication in PLOS ONE. Congratulations! Your manuscript is now being handed over to our production team.

Kind regards, 

on behalf of

Dr. Mario Milazzo 

Academic Editor

PLOS ONE